# Recent Advances in Polymer-Based Vaginal Drug Delivery Systems

**DOI:** 10.3390/pharmaceutics13060884

**Published:** 2021-06-15

**Authors:** Tomasz Osmałek, Anna Froelich, Barbara Jadach, Adam Tatarek, Piotr Gadziński, Aleksandra Falana, Kinga Gralińska, Michał Ekert, Vinam Puri, Joanna Wrotyńska-Barczyńska, Bozena Michniak-Kohn

**Affiliations:** 1Chair and Department of Pharmaceutical Technology, Poznan University of Medical Sciences, 60-780 Poznań, Poland; froelich@ump.edu.pl (A.F.); bajadach@ump.edu.pl (B.J.); adamtatarek97@gmail.com (A.T.); piotr.gadzinski@gmail.com (P.G.); falana.aleksandra@gmail.com (A.F.); kingagralinska@gmail.com (K.G.); shhiraan04@gmail.com (M.E.); 2Department of Pharmaceutics, William Levine Hall, Ernest Mario School of Pharmacy, Rutgers, The State University of New Jersey, Life Sciences Building, New Jersey Center for Biomaterials, Piscataway, NJ 08854, USA; vp239@dls.rutgers.edu (V.P.); michniak@pharmacy.rutgers.edu (B.M.-K.); 3Division of Infertility and Reproductive Endocrinology, Department of Gynecology, Obstetrics and Gynecological Oncology, Poznan University of Medical Sciences, 33 Polna St., 60-535 Poznań, Poland; jwrotynska.barczynska@gmail.com

**Keywords:** vagina, polymers, mucoadhesion, drug delivery systems, nanoparticles, gels, films, patches

## Abstract

The vagina has been considered a potential drug administration route for centuries. Most of the currently marketed and investigated vaginal formulations are composed with the use of natural or synthetic polymers having different functions in the product. The vaginal route is usually investigated as an administration site for topically acting active ingredients; however, the anatomical and physiological features of the vagina make it suitable also for drug systemic absorption. In this review, the most important natural and synthetic polymers used in vaginal products are summarized and described, with special attention paid to the properties important in terms of vaginal application. Moreover, the current knowledge on the commonly applied and innovative dosage forms designed for vaginal administration was presented. The aim of this work was to highlight the most recent research directions and indicate challenges related to vaginal drug administrations. As revealed in the literature overview, intravaginal products still gain enormous scientific attention, and novel polymers and formulations are still explored. However, there are research areas that require more extensive studies in order to provide the safety of novel vaginal products.

## 1. Introduction

The first known data concerning the vaginal application of therapeutics appeared in medical books from Ancient Egypt. The Kahun Medical Papyrus (dating from 2250 years B.C.) is considered to be the oldest known gynecological handbook. Among others, it contained specific information on treating vaginal inflammations and described various methods of contraception [1]. Interestingly, some of the treatment methods proposed back then may now be considered controversial and even risky. One of the most interesting formulations described in the ancient documents were contraceptive suppositories, composed of crocodile manure mixed with honey and sodium carbonate. Another ancient source of vaginal prescriptions is the Papyrus Ebers (about 1550 B.C.), which describes contraceptive tampons made of lint and soaked in honey and acacia tips. Acacia shrub was used as a popular source of gum arabic. After the insertion of a tampon, the gum released lactic acid which acidified the vagina [2,3,4]. Nowadays, lactic acid is a well-known and very common spermicidal agent used in modern contraceptive gels and creams [5,6]. It is also noteworthy that in the 15th century, people already had the knowledge that some substances are able to penetrate to the systemic circulation after vaginal administration. A quite common, however inglorious practice, was the vaginal application of arsenic or other poisonous substances in order to induce abortion or commit suicide [7].

At present, vaginal formulations are mostly used to deliver topically acting drugs such as antimicrobials [8,9,10,11,12], spermicides [13,14], antimycotics [15] or to introduce drugs to the systemic circulation [16,17,18], mostly for hormonal therapy or contraception [19,20]. Much interest is also focused on the vaginal delivery of microbicides in order to inhibit sexual transmission of HIV [21,22], HPV [23,24,25] or HSV [26,27,28].

The abundance of blood vessels in the vagina is a definite advantage, especially in relation to systemic formulations [29,30,31,32]. The primary benefit is the fact that active substances do not undergo the first-pass hepatic metabolism. Furthermore, the vaginal epithelium has a very suitable permeability, even in the case of drugs with a high molecular weight such as peptides or proteins [33,34,35].

When compared to the oral route, there are several benefits of vaginal administration. These include, for example, reduction in the dose, less frequent dosing, reduced sideeffects, no hepatic first-pass effect. However, some challenges have to be overcome, including dilution by vaginal fluid or peristaltic activity of the vaginal wall [36].

Among the known vaginal malfunctions, the most common include bacterial vaginosis, aerobic vaginitis, candidiasis, sexually transmitted infections, atrophic vaginitis, desquamative inflammatory vaginitis, cervicitis, and mucoid ectopy. The symptoms related to them are usually non-specific such as itching, burning, pain, abnormal bleeding, or discharge. However, most of the mentioned can also be associated with vaginal dermatoses, allergic and irritant reactions [37].

According to data given by the Centers for Disease Control and Prevention, bacterial vaginosis (BV) is the most common vaginal infection among women betweenthe age of 15 and 44. It is estimated that in the United States, the prevalence of women who suffered from BV reached about 21.2 million in 2004 [38,39,40]. Among vaginal infections, candidiasis is the second most common. Other vaginal diseases include vaginal or vulvar cancers. In general, they are regarded as rare, with 0.7 per 100,000 diagnosed with vaginal and 2.6 per 100,000 with vulvar in the year 2017.

To properly design a vaginal formulation, it has to be taken into account that the conditions in the vaginal cavity are very unstable. The acidity, temperature, production of vaginal fluid, and thickness of the epithelium strongly depend on the phase of the menstrual cycle, sexual activity, age of the patient, and concomitant diseases [41,42,43,44].

Most vaginal drug delivery systems contain various types of natural or synthetic polymers. Their main role is to assure the contact of the drug with the site of action as long as it is possible and provide a controlled drug release, repeatedly and in a predictive manner. A large variety of vaginal drug delivery systems are already in use or under investigation. These include gels, creams, foams, tablets, capsules, suppositories, pellets, microparticles, nanoparticles, patches, films, or rings. The aim of the presented paper was to prepare acomprehensive review of the latest developments in the field of polymer-based vaginal drug delivery systems, indicate the most extensively investigated research directions, and discuss future trends and possible obstacles in the studies focusing on vaginal drug delivery.

## 2. Anatomy and Physiology of the Vagina

The human vagina is a fibromuscular tubular organ that connects the vulva, cervix, and uterus, and the organs of the upper reproductive tract [45]. It is divided into two parts, the upper and lower, which originate from the mesoderm and epiderm, respectively [46]. The vaginal wall is composed of three layers: *i.* epithelium, which is the inner mucosal layer (*tunica mucosa*) and consists mainly of non-keratinized squamous cells, *ii.* muscle layer (*tunica muscularis*) built of smooth muscle cells and *iii.* external (outer) membrane filled with collagen [46,47]. The inner surface of the vagina is built of transverse folds called wrinkles (lat. *Rugae vaginales*). They are responsible for maintaining the correct tension and stiffness of the whole organ. The thickness of the epithelium changes during the menstrual cycle in the range of 200–300 microns. Its properties depend mainly on the level of sex hormones. Estrogens produced during the proliferative phase induce the growth and cornification of the epithelial layer [48,49,50]. During pregnancy, an epithelial cell layer is thin. In the menstrual cycle, the concentration of enzymes from the group of aminopeptidases and endopeptidases in the vaginal fluid changes [51,52]. This may create a problem with the selection of thecorrect dose of the active substance. Drug release, distribution, and absorption after vaginal administration strongly depend on the amount and the properties of the vaginal fluid. The fluid usually contains components from vaginal wall transudate, cervical and vestibular glands secretions, exfoliated epithelium cells or residual urine, and fluids from the upper reproductive tract [53]. According to the extensive data presented by *Owen and Katz* [54], the organic compounds identified in the vaginal fluid include proteins, carbohydrates, and other small molecules such as urea, glucose or lactic and acetic acid, and others. The fluid also contains inorganic ions such as potassium, sodium, calcium, or chloride. However, it must be taken into account that the vaginal secretions vary according to the actual conditions, menstrual cycle phase, and existing diseases [55,56]. In physiological conditions, the fluid is usually thick, clear, or slightly opaque but can turn to creamy, clumpy, green, or yellow with a specific odor as a result of a bacterial, fungal, or other infection [57,58]. The physiological amount of the fluid is relatively small (usually between 0.5 and 1.0 mL). Nevertheless, it may dissolve or liquefy the formulation and have a negative influence on the formulation’s residence time or accelerate the disintegration resulting in weakening of the therapeutic activity. In the case of semisolid preparations, the fluid can weaken interactions within the polymer matrix and lead to a change of the rheological properties. Other factors that are crucial in drug release and distribution are pH and osmolarity of the vaginal environment [54], especially in the case of polymers with pH-sensitive moieties.

Physiological vaginal bacterial flora is a complex system. Its composition changes dynamically depending on many factors. The probiotic microbe that predominates in the vaginal environment is the anaerobic, rod-shaped lactic acid bacteria *Lactobacillus acidophilus*. It produces lactic acid, which maintains vaginal pH in the range of 3.5–4.5 [53,59,60,61]. Besides low pH, several other mechanisms have been developed to prevent the colonization of the vagina by exogenous pathogens. These include the production of antimicrobial bacteriocins, peroxidases, and other organic acids. Moreover, the vaginal wall epithelial cells constantly exfoliate, which results in the efficient removal of the microbes [53].

One of the most important factors that havea significant influence on the efficiency of the vaginal formulations is the adhesion to the mucous membrane. The mucoadhesive properties provide a longer residence time and a more intimate contact with the site of action. In the case of systemic delivery, the properly designed mucoadhesive systems are able to decrease the distribution of active pharmaceutical ingredients (API) throughout the vaginal cavity and promote its penetration into the systemic circulation [62].

The vaginal mucous membrane consists mainly of water, inorganic salts, carbohydrates, lipids, salts, DNA, enzymes, and mucins. Its features resemble the properties of hydrogels [32,63,64]. Mucin belongs to glycoproteins and is responsible for the formation of a hydrogel layer on the external mucosa [62]. Mucoadhesion can be defined as the phenomenon of adhesion and bonding of the polymers with the vaginal mucosa through physical and chemical interactions [65]. Connection and formation of mucoadhesive interactions can be divided into three stages: *i*. wetting and swelling of the polymer, *ii*. connection of the polymer chains with the chains of mucin, *iii*. formation of weak chemical bonds. Generally, it can be regarded that the greater the mucoadhesion ability of the polymer is, the longer the residence time of the drug formulation in the application site gets. Mucoadhesion of vaginal formulations depends mainly on the physicochemical properties of the polymer, including its molecular weight, degree of ionization, and functional group types [66]. Mucoadhesive polymers are divided into three groups: 1. polymers that reveal adhesive properties after being placed in a water environment; 2. polymers that connect with mucous membranes by non-specific binding such as non-covalent or electrostatic; 3. polymers with the ability to bind to specific receptors on the cells or surface of the mucosa [67].

Once the formulation reaches the vaginal mucosa, it can release the drug into the vaginal cavity or on the surface of the vaginal mucosa. Depending on the type of formulation and properties of the active compound, systemic activity can also be achieved as a result of permeation to the vaginal vascular system [68,69]. According to the studies presented by van der Bijl &van Eyk [70], the vaginal mucosa reveals a very similar permeability to water as buccal mucosa, but in comparison to the intestinal and colonic mucosa, its permeability is higher. In addition, it is more permeable for various drugs than intestinal and colonic mucosae, even in the case of compounds with amolecular weight higher than 300 kDa [70,71,72]. A scheme depicting the vaginal route is presented in Figure 1.

## 3. Polymers Used in Vaginal Drug Delivery Systems

### 3.1. Polymers of Natural Origin

#### 3.1.1. Polymers from Plant Sources

##### Cellulose and Its Derivatives

Cellulose is regarded as the most abundant organic compound throughout the world [73]. It is an unbranched polysaccharide, built of 3000–14,000 glucose molecules linked by linear β-1,4-glycosidic bonds. It is the scaffold component of cell walls and tissues in most plants [74]. Pure and unmodified cellulose is insoluble in water and most organic solvents [75]. In order to obtain solubility and achieve swelling properties, the hydroxyl groups of the main backbone are subjected to esterification or etherification. The semisynthetic cellulose derivatives constitute a large and diverse group of compounds differing in terms of polarity, water solubility, swelling properties, and thus possible pharmaceutical and biomedical applications [76]. The most commonly used cellulose derivatives are microcrystalline cellulose (MCC), methylcellulose (MC), ethylcellulose (EC), hydroxyethyl cellulose (HEC), hydroxypropyl cellulose (HPC), hydroxypropyl methyl cellulose (HPMC), and sodium carboxymethyl cellulose (Na-CMC) [77,78].

Depending on the unique and varied properties, semisynthetic celluloses are widely used both in oral and topical drug delivery. In the case of tablet technology, their most popular applications include their use as binders, compressibility enhancers, fillers, diluents. Depending on the way of interacting with water, celluloses may be used for the modification of drug release by acting as disintegrants, matrix-forming components, or coating agents. Due to their swelling ability, they can also be used as thickeners and stabilizing agents in liquid and semisolid dosage forms. Because of the mucoadhesive properties, semisynthetic celluloses are often investigated as components of vaginal drug delivery systems, mostly in gels or viscous liquids but also tablets and micro- or nanoparticulate formulations [79,80,81].

Pectin

Pectin is usually defined as a diverse and most complex group of oligosaccharides and polysaccharides abundantly occurring in plant cell walls. Its main component is the esterified D-galacturonic chain [82]. In the case of natural pectin, the acid groups are esterified with methoxy residues. The free hydroxyl groups can also occur in the acetylated form, and additionally, the galacturonic acid main chain can be substituted with rhamnose groups [83]. The presence of the latter disrupts the chain helix formation [84]. Depending on the source, pectin can also contain other xylose, galactose, or arabinose residues, located in side chains. Pectins are differentiated mainly due to methoxy group content and classified as high methoxy- (>50% esterified) and low methoxypectins (<50% esterified) [85]. It is noteworthy that this model may vary significantly in terms of particular domains, e.g., chain length, sugar composition, and the degree of methylation or acetylation [86]. Among the most important features of pectin, its resistance to the acidic environment is most beneficial in oral drug delivery with a modified release in the lower gastrointestinal tract. The gelation mechanism strongly depends on the methylation degree of the main linear structural element of the polymer. In the case of high methoxyl group content, a gel is formed at pH < 3.5, usually in thepresence of an additional substance (e.g., sucrose), decreasing water molecules activity [86]. In vaginal drug delivery studies, pectins are mainly investigated as mucoadhesive components of various formulations. Their pH-dependent behavior can also strongly influence the drug release mechanism [87].

##### Alginates

Alginates are biocompatible and biodegradable anionic polysaccharides occurring naturally in brown seaweeds (*Phaeophyceae*) [88]. Among the whole group, sodium alginate is the one most commonly used for pharmaceutical and biomedical purposes [89]. Alginic acid is a copolymer containing D-mannuronic and L-guluronic acids organized in blocks separated with sequences of the same units organized randomly [90]. The exact composition of the compound depends on its source of origin. Alginates reveal a high water binding capacity due to their hydrophilic nature [91]. In the presence of divalent and multivalent cations, alginate solution undergoes an ionotropic gelation process [92,93]. The properties of the obtained gel depend mostly on the composition of the polymer. Gel strength is higher in the case of compounds with a higher guluronic acid content [92]. Alginates can be applied as thickeners and stabilizers in liquid and semisolid pharmaceutical formulations [94]. They are also investigated as binders and hydrophilic matrix-forming agents in prolonged-release solid dosage forms [95]. Moreover, the ability to undergo ionotropic gelation may be advantageous in terms of in situ gelling system formulations [96,97].

Starch

Starch is one of the most abundant plant polysaccharides and the main carbohydrate in the human diet [98]. In fact, it consists of two compounds: linear amylose (25%) and branched amylopectin (75%), both composed of multiple α-D-glucose units. In the first one, structural elements are linked with α-1,4 bonds, while in the other one, α-1,4, α-1,3, andα-1,6 bonds are observed [99,100]. Starch is insoluble in cold water, while at higher temperatures reveals a tendency to swell and form gels [101,102]. For vaginal drug delivery purposes, starch, as well as its derivatives, are investigated mainly as the component of tablets, micro- or nanoparticles, gels, etc. [103,104,105,106].

Carrageenans

Carrageenans belong to a family of linear polysaccharides obtained from red seaweeds by alkaline extraction. The polymer chain is formed by repeating disaccharide units of alternating 3-linked β-D-galactopyranose, 4-linked α-D-galactopyranose or 4-linked α-D-galactopyranose or 4-linked 3,6-anhydro-α-D-galactopyranose [107]. Thereare three types of carrageenan mostly known for their application in vaginal formulations technology, which includes iota-carrageenan (ι-carrageenan), kappa-carrageenan (κ-carrageenan), and lambda-carrageenan (λ-carrageenan) [108]. Each type differs both in structure and the content of ester sulfate [109].According to specific and unique properties, carrageenans can be used both as excipients or active components of vaginal formulations. In the first case, they are mostly used as mucoadhesive additives or structure-forming agents in solid and semisolid formulations [110,111]. Moreover, carrageenans have the potency to inhibit microbial and viral infections, which is considered as their prevalence over other natural polymers in terms of transmission of vaginal infections [112,113].

#### 3.1.2. Polymers Derived from Animal Sources

##### Chitosan

Chitosan is obtained as a product of deacetylation of chitin, a natural component of numerous invertebrate exoskeletons [114]. The reaction is usually performed in the presence of concentrated NaOH; however, the process may also be conducted with the use of a chitin deacetylase [115]. The most extensively exploited source of chitin used as a substrate to obtain chitosan isedible crustaceans such as shrimps and crabs [116,117]. Chitosan is a group of linear copolymers consisting of glucosamine and N-acetylglucosamine connected with β-1,4 bonds [118]. It is important to notice that chitosan differs significantly from its substrate in terms of solubility. Chitosan is soluble in acidic solutions thatarea result of the presence of free amine groups in the molecule. In the case of chitin, amine groups are mostly acetylated, which makes the polymer practically insoluble in an acidic and slightly alkaline environment [119].

Because of its interesting properties, chitosan is extensively investigated both as an active ingredient and as an excipient [120]. It is non-toxic, biodegradable, and biocompatible [121]; therefore, it can be safely applied as a component of pharmaceutical formulations [122,123]. Moreover, some antibacterial properties [124] and a hemostatic activity of chitosan were revealed [125]. Orally administered chitosan can effectively decrease the serum cholesterol level, which can be useful in hypercholesterolemia therapy and arteriosclerosis prevention [126]. It is also noteworthy that the polymer presents excellent mucoadhesive properties [127] and a gelling ability that is favorable in terms of topical formulation design [128,129]. Another interesting feature of chitosan related to the mucoadhesive characteristics is its ability to accelerate the wound-healing process. This property is related to the gradual depolymerization of chitosan. The products released in this reaction improve the organization of novel collagen fibers and enhance the production of hyaluronic acid [130,131].

##### Hyaluronic Acid

Hyaluronic acid is a glycosaminoglycan occurring naturally in the human body [132]. As a component of the extracellular and pericellular matrix, it is present in nearly all tissues of human and other vertebrates’ bodies [133]. It is most commonly known as an important component of synovial fluid [134]. Hyaluronic acid molecules consist of repeating units composed of D-glucuronic acid and N-acetyl-D-glucosamine connected by alternating β-1,4 and β-1,3 bonds. It is noteworthy that hyaluronic acid molecules are large, unbranched, with molar masses reaching even 10^7^ Da [135]. They form double helices connected to each other as a result of interactions between hydrophobic areas formed by axial hydrogen atoms of CH groups connected into β-sheets. Adjacent planar structures are linked with hydrogen bonds into a three-dimensional network [136]. The physicochemical properties of hyaluronic acid strongly depend on its molecular weight and the concentration of the solution. Low molecular weight hyaluronic acid solutions reveal Newtonian properties while systems composed of large molecules are non-Newtonian, showing clearly the viscoelastic properties. The same tendency is observed when diluted and concentrated solutions are compared. Moreover, the rheological characteristics of hyaluronic acid solutions depend also on the pH, which is related to the presence of carboxylic groups in its molecule [136,137]. Hyaluronic acid has many important functions in the human body. Because of its high polarity, it can bind high amounts of water, which is important in terms of proper skin functioning [138]. Moreover, hyaluronic acid also serves as a scaffold for proteins and cells, affecting their proliferation and tissue regeneration [139]. During tissue damage and infections, hyaluronic acid is rapidly degraded into simple sugars. It was also found that the physiological activity of the polymer depended on its molecular weight. Large molecules usually act as matrix components and reveal immunosuppressive and anti-angiogenic properties. Medium-sized molecules are considered angiogenic and immunostimulatory agents, while small hyaluronic acid molecules play important roles in signal transduction through different pathways [135]. Currently, hyaluronic acid is widely applied in the medicine, pharmacy, and cosmetic industry. The most important medicinal areas employing hyaluronic acid are ophthalmic and plastic surgery, ophthalmology and wound healing. In surgical procedures, it is used as a filler to create operating space. Moreover, it can be applied as intra-articular injections in rheumatoid arthritis. According to Greenberg et al., hyaluronic acid improves viscoelastic properties of synovial fluid impaired in a degenerative process, inhibits further cartilage degradation, acts as an inflammatory agent and also exerts some analgesic effect [140]. Exceptionally high concentrations of hyaluronic acid are observed during inflammations of the vagina caused by the fungi (recurrent vulvovaginal candidiasis; RVVC) [141].

The first applications of hyaluronic acid were related to accelerated wound healing and skin recovery. Today, it is widely used in the cosmetic and pharmaceutical industries. It is a component of eye drops, intra-articular injectable solutions, lotions for bladder irrigation, aerosols used to treat asthma, solutions for mouthwash, and anti-acidic formulations. The potential use of hyaluronic acid in the prevention of viral infections is also investigated [142]. In the process of developing a formulation containing hyaluronic acid, it should be taken into account that it is degraded by hyaluronidase and hydroxyl radicals [143].

##### Gelatin

Gelatin is a natural biopolymer obtained from animal cartilage and bones as a result of collagen hydrolysis. The properties of the obtained product depend on the technological process employed to pretreat the animal material before collagen extraction. In alkaline conditions, amide groups of asparagine and glutamine are transformed into free carboxylic groups, while in the acidic process, these groups remain intact. As a result, products revealing different electrical properties are obtained. Gelatin produced at alkaline conditions contains more acidic groups that may undergo an ionization process in solution. Negative charge localized along the polymer chain results in the lowering of the isoelectric point when compared to the product obtained in acidic conditions. It is noteworthy that gelatin produced with the use of the method involving acidic hydrolysis is similar to the untreated collagen [144]. Gelatin reveals many advantageous properties important in pharmaceutical technology, e.g., excellent biocompatibility and biodegradability. Therefore, it is widely applied as a component of soft and hard capsule shells marketed all over the world [145]. Moreover, it has also been used in commercial formulations as a plasma expander (Gelofusin^®^, B. Braun Medical Ltd., Sheffield, UK), as a hemostatic agent for wound closure (Gelita^®^, B. Braun Medical Ltd., UK), and for impregnation of polyester, implants applied in reconstructive surgical procedures in the aorta and peripheral arteries (Uni-Graft^®^K DV, B. Braun Medical Ltd., UK) [146]. It is also widely used as a stabilizer in live attenuated viral vaccines; however, allergic reactions related to gelatin presence have been reported [147].

#### 3.1.3. Microbial Polymers

##### Gellan Gum

Gellan is an anionic polysaccharide secreted by *Sphingomonas* (formerly *Pseudomonas*) *elodea* bacteria as a product of the fermentation process. Gellan molecules are linear and consist of repeating tetrasaccharide units containing l-rhamnose, d-glucose, and d-glucuronic acid moieties in the molar ratio 1:2:1. In the native form gellan main backbone is substituted with acetyl and L-glyceryl moieties, which can be removed in a hydrolysis process leading to obtaining the low-acetyl gellan gum. Both forms are commercially available [148]. The most important feature of gellan gum is its ability to form gels in the presence of mono-, di-, and trivalent cations, which can form coordinate bonds with carboxylic groups of the polymer and stabilize the three-dimensional structure. The properties of the obtained physical gels depend on the acetylation degree of the polymer. In the case of the low-acetylated form, the rigid and brittle product is obtained, while in the case of high-acetylated gellan soft semisolid gels are observed. The described feature is favorable in terms of in situ gelling systems forming upon contact with physiological fluids containing mentioned cations.

##### Xanthan Gum

Xanthan gum is a microbial polysaccharide obtained in a fermentation process of cabbage plant bacterium *Xanthomonas campestris*. The biopolymer has been produced industrially since 1964, and in the late 1960s, it was granted the approval of the FDA as a food additive. Currently, it is employed in several areas, e.g., the food industry, personal care products and pharmaceutics. It may act as a stabilizer in disperse systems such as emulsions and suspensions, and because of excellent swelling properties and a shear-thinning behavior, it may be used as a thickening agent in topical drug dosage forms and cosmetics, as well as a structure enhancing additive in food products [149,150].

The most important structural element of the xanthan molecule is a backbone consisting of glucose moieties connected with β-1,4-glycosidic bonds also observed in cellulose molecules. The cellulose backbone is connected to side chains consisting of two mannose and one glucuronic acid moieties. The side chains are attached to the main structural element through β-1,3-glycosidic bonds, and some of them are terminated with a pyruvic acid residue. Moreover, the hydroxyl group in position 6 of one or both mannose moieties may be esterified with acetic acid [151]. The conformation of xanthan molecules depends on the temperature.

At physiological conditions, they occur in the form of a helix, while at higher temperatures, the transition into a disordered state is observed. The same process can also be induced by dilution [152].

One of the most important properties of xanthan is its ability to swell and form physical gels revealing shear-thinning properties. The obtained solutions are stable in a wide range of environmental conditions, such as pH, temperature, and ionic strength [151]. Therefore, this polymer is widely investigated as a thickening agent in topical dosage forms, including dermal and ocular formulations. Moreover, swelling properties are advantageous in terms of prolonged release in oral drug delivery, and xanthan is studied as a hydrophilic carrier in matrix tablets.

### 3.2. Synthetic Polymers

#### 3.2.1. Poloxamers

Poloxamers are synthetic block copolymers composed of hydrophobic poly(propylene oxide) (PPO) units with two hydrophilic blocks of poly(ethylene oxide) (PEO). The building blocks of poloxamer reveal different polarities, and the presence of both elements in the molecule makes it amphiphilic. The hydrophilic-lipophilic balance (HLB) value characterizing amphiphilic properties of poloxamer depends on the molar ratio of propylene oxide and ethylene oxide blocks [153]. The most important representative of this chemical class is poloxamer 407, registered as Kolliphor^®^ P 407, previously known also as Pluronic^®^ F 127 (BASF, Florham Park, NJ, USA) and Synperonic™ PE/F 127 (Croda Health Care, Plainsboro, NJ, USA). It is usually applied as a thickening agent in liquid and semisolid formulations and a solubilizing agent [154,155]. Another polymer frequently applied in pharmaceutical formulations is poloxamer 188. Both compounds differ from each other in terms of molecular weight, which is 4000 and 1800 for poloxamer 407 and 188, respectively. Another difference is EO blocks content, which is 70% in poloxamer 407 and 80% in poloxamer 188. These two parameters are crucial for the physicochemical properties of poloxamers. Their solubility in water increases with the contentof more hydrophilic ethylene oxide units and decreases at higher molecular weights [156]. One of the most important properties of poloxamer is its thermosensitivity. It was demonstrated that the increase in temperature resulted in the increase in the viscosity and the transformation from a liquid system into a semisolid one [157]. As the temperature increases polarity of more hydrophobic PPO blocks decreases that results in their further dehydration. In these conditions, poloxamer molecules form spherical micelles with a hydrophobic core composed of PPO units and a hydrophilic shell built of PEO chains. With a further increase in the temperature, micelles organize into a three-dimensional network, which is related to gelation. This process is fully reversible and takes place at a certain temperature, depending on the polymer concentration, molecular weight, and structure [158]. There are numerous studies describing pharmaceutical formulations with poloxamer composed in a way enabling gelation at physiological temperature [159,160,161,162,163,164,165]. These systems remain liquid at room temperature, which provides an easy application, for example, as an injection, rectal formulation, or ophthalmic drops. At the application site, they transform into gels that may provide a prolonged drug release or increase the residence time in the case of formulations applied rectally or vaginally. However, the lack of mucoadhesive properties is a major drawback of these polymers but can be improved by the addition of another mucoadhesive material [166].

#### 3.2.2. Polyacrylates

Polyacrylates are a group of cationic and anionic synthetic esters of acrylic and methacrylic acids, with different structures and physicochemical properties. They are available in different forms (powders, granules, organic solutions, aqueous dispersions) and commercial products such as Eudragit^®^ (Evonik Industries AG, Darmstadt, Germany), Kollicoat^®^ (BASF, Florham Park, NJ, USA), Eudispert^®^ (Röhm Pharma, Darmstadt, Germany). Specific polyacrylates differ in terms of their abilities of dissolution and swelling. Depending on the structure, they may form coating films with different solubilities at different pH values [167,168]. Mostly they are used as film-forming and coating components in the preparation of tablets, enteric-coated capsules, and oral dosage forms with a modified release. Polyacrylates can also be used to form the basis for semisolid, transdermal, vaginal, and rectal drug delivery systems. A copolymer of methacrylic acid with methyl ester of this acid (Eudispert^®^) is especially useful in these kinds of formulations because of its bioadhesive properties [168,169,170,171,172].

A special group of polyacrylates isCarbopols^®^. They are synthetic polymers of acrylic acid. They differ in molecular weight, number and type of crosslinks, and properties, especially viscosity and bioadhesion. The main backbone of these polymers is formed by acrylic acid residues, and its adjacent chains may be cross-linked by an allyl ether radical of sucrose or by pentaerythritol. The carboxyl groups comprise 52 to 68% of weight [173,174].

Carbopols are used as excipients in pharmacy. They play an important role as emulsifiers and emulsion stabilizers (e.g., Pemulen^TM^) [175]. They are also used as release-modifying agents, hydrophilizing substances, binders, and viscosity modifiers. Due to the wide range of useful properties, Carbopols have been employed in numerous drug delivery systems. Aconcentration of 5–15% is used in the formation of capsules and tablets. They are excipients in liquid and semisolid formulations, for example, creams, gels, enemas, lotions, ointments for topical use, rectal and vaginal drugs. Carbopols are also tested for their application in multiple oral drug delivery systems and in the oral mucoadhesive systems with controlled release [174,176,177,178,179,180].

#### 3.2.3. Polyvinylpyrrolidone

Povidone is a synthetic polymer formed by the linearly arranged 1-vinyl-2-pyrrolidone. The degree of polymerization determines the molecular weight of the compound, which influences the properties of polyvinylpyrrolidone [181]. Higher molecular weight causes an increase in viscosity and a decrease in solubility of this substance. Povidone is a white, hygroscopic powder with no specific odor, and it is easily soluble in water and many organic solvents. After oral application, it is not absorbed from the gastrointestinal tract and, when applied to the skin, does not cause irritation or sensitization. The polymer has been used as a stabilizer and thickener of the suspensions and solutions used orally and topically. It is also used as a solubilizer in oral and parenteral formulations, as a binding agent in the wet granulation process, and as a disintegrant in tablet technology [182,183].

#### 3.2.4. Polyethylene Glycol

Polyethylene glycols are also known as macrogols. Depending on the degree of polymerization, they differ in molecular weight and consistency. With molecular weight increase, the increase in the viscosity may be observed, and the physical form can range from liquid to a hard wax. All kinds of polyethylene glycols are soluble in water and miscible with each other. They are stable, hydrophilic substances exhibiting no skin irritation. Furthermore, their removal from the skin is very easy, so they are often used as an ointment base and to prepare suppositories. Macrogols have also been applied in ocular and oral formulations, as well as injections [184,185,186].

## 4. The Examples of Polymer-Based Vaginal Formulations

Due to the fact that the drug delivered through the vaginal route has to overcome physiological and anatomical barriers to achieve a localized action or reach the systemic circulation, it is important for the drug form to have specific properties. Those are, for example, a long residence time at the site of action, proper parameters of drug release, and adhesion to the mucosa. Mucoadhesive formulations based on the appropriate polymers are particularly interesting and seem to be the most promising in the case of vaginal drug delivery. This group comprises conventional drug forms, e.g., ointments, creams, tablets, and suppositories, as well as newly discovered rings, nanoparticles, or films [187]. All listed drug forms are meant to optimize the action of the drugs administered into the vagina and will be described in detail in this chapter. The dosage form classification applied in this review is presented in Figure 2.

### 4.1. Semisolid Formulations

Among the most popular semisolid formulations intended for vaginal use, gels, creams, and less frequent ointments are mentioned [32,69,188,189]. The advantages of these systems comprise ease of application, high acceptability, and low production cost. However, semisolid systems are also considered problematic in terms of possible leaks, discomfort after application, messiness, and short residence time at the administration time, which might also contribute to the limited efficacy. In order to improve the therapeutic effect observed as a result of vaginal semisolid products, the therapy sometimes requires frequent administration, which might be considered an inconvenience [188,190]. Another approach involves the application of mucoadhesive polymers responsible for enhancing interactions with mucous membrane and increasing the residence time in the vagina. Some studies indicate the possibility of employing environmentally sensitive polymers increasing the viscosity of the formulation after the administration. In this way, the product is transforming from liquid to semisolid form upon contact with vaginal conditions. An important issue related to the application and effectiveness of vaginal products is acceptability by patients. It is noteworthy that semisolid dosage forms are considered by the patients as convenient in terms of application and are more likely to be preferred over vaginal rings, vaginal suppositories, and vaginal tablets [189,191]. However, it is noteworthy that patients’ acceptance is a complex issue and may depend on many cultural, socioeconomic and other factors [188,192]. Recently, one of the most frequently investigated areas in vaginal drug delivery is related to the formulation of antiviral agents-loaded products as potential preventive products in HIV and sexuallytransmitted infections (STIs). It was shown that in these products, the most important factor affecting patients’ preferences and choices is dosing frequency [193]. In this aspect as the biggest drawback of semisolids their poor retention time is mentioned [194].

#### 4.1.1. Gels

Even though gels frequently occur in numerous scientific and non-scientific areas, the definition of the gel is difficult, and there are several different approaches to the description of these systems. According to Almdal et al. [195], gels are defined as soft, solid-like, or solid material consisting of two or more components. One of the components is liquid and occurs in a significantly higher amount than the other one. The proposed definition also emphasizes the specific rheological properties of gels, which is storage modulus (G’) significantly higher than loss modulus (G”) and exhibiting prolonged plateau. It is also noteworthy that heterogenous materials are excluded from this definition. According to Rogovina et al. [196], the gel is an elastic solid containing two or more components. One of the components is liquid, and the other one forms a three-dimensional network. The type of bonds occurring in the network determines gel type. Chemical gels are bonded with strong covalent bonds, while in physical ones, mostly hydrogen bonds are present. The network-forming component is usually a polymer; however, there is also a possibility to obtain gel with the use of low molecular weight gelators [197]. Considering the type of liquid component, gels can be classified as hydrogels when they are water-based or organogels when the liquid component is non-aqueous. The general classification of gels according to different criteria is depicted in Figure 3. It is noteworthy that gels applied in vaginal drug delivery are mostly weak physical hydrogels obtained with polymer gelling agents.

Most of the investigated gel-based dosage forms aim at the delivery of antimicrobial agents in various vaginal infections, including fungal [11,36,198,199] and bacterial ones [12,199,200,201]. However, numerous studies focus on the effective delivery of contraceptive agents and the prevention of HIV transmission [202]. It is important to note that the safety and efficacy of several gel formulations containing mainly dapivirine and tenofovir were subjected to clinical trials [203]. Taking into consideration the most extensively investigated areas in terms of pharmaceutical technology and optimization of drug carriers, mucoadhesive and thermosensitive systems should be mentioned.

#### 4.1.2. Mucoadhesive Drug Delivery Systems

Mucoadhesion occurs as a result of interaction between formulation components and vaginal mucous membrane or mucus layer lining its surface. The most important elements of mucus are mucins, water-soluble glycoproteins revealing a high degree of glycosylation. The subunits of mucin are connected with disulfide bridges and form large three-dimensional gel structures [31,204]. Components of mucoadhesive formulations interact with mucin through hydrogen bonds, van der Waals, or electrostatic interactions. Therefore, the most important feature of potentially mucoadhesive polymers is the presence of a large number of functional moieties, such as hydroxyl or carboxyl, sulfate, and amine groups. Other features positive or negative charges in the molecule, chain flexibility enabling interpenetration of polymer and mucin lattices, and favorable surface properties, allowing for spreading the formulation on the mucous membrane [205].

One of the most commonly used bioadhesive polymers is chitosan, a cationic linear polysaccharide obtained as a product of chitin deacetylation, also known for antimicrobial [206] and wound-healing properties [130]. Bonferoni et al. [207] investigated two types of chitosan differing in molecular weight as carriers for mucoadhesive gels designed for the controlled release of lactic acid. Taking into consideration the differences in the active ingredient release in different media, it was assumed that the release of lactic acid occurred as a result of diffusion and ionic displacement. It was also shown that the lower molecular weight of the polymer was associated with stronger bioadhesive interactions. The same research group [208] analyzed gels obtained with chitosan citrate, which was supposed to chelate calcium cations participating in the regulation of gap and tight junctions. As active ingredients, acyclovir and ciprofloxacin hydrochloride were used. The applied chitosan derivative was also tested for its activity toward proteolytic enzymes, carboxypeptidase A and leucine aminopeptidase. Inhibition of these enzymes is considered advantageous in terms of delivery of hydrophilic and macromolecular compounds through the mucous membrane. Even though the obtained permeation results were satisfactory when compared to controls, the comparison with chitosan hydrochloride did not show any statistically significant differences. Senyiğit et al. [209] analyzed chitosan-based gels with miconazole and econazole nitrates. In the study, the effects of polymer molecular weight on the active ingredients release, as well as vaginal retention and mucoadhesive properties, were investigated. Moreover, the antimicrobial properties of all formulations presented in the study were evaluated. It was shown that the formulation prepared with medium molecular weight polymer revealed the best properties in terms of vaginal drug delivery. Tuğcu-Demiröz et al. [210] presented comparative studies focused on different polymer gels designed for the systemic delivery of oxybutynin, an antimuscarinic agent applied in overactive bladder. As thickening agents, chitosan, hydroxypropyl methylcellulose (HPMC K100M), and poloxamer 407 were applied. The best cohesiveness and mucoadhesion were observed in the case of HPMC K100M. Moreover, the performance of all semisolid formulations in vivo was compared to the marketed product administered in tablets. It was shown that HPMC-based gel could be a suitable alternative to the oral formulation. Cevher et al. [211] investigated hydrogels containing chitosan and polycarbophil covalently modified with thioglycolic acid and cysteine, respectively. The obtained products were applied as carriers for clomiphene citrate for potential therapeutic application in human papilloma virus (HPV) infections. It was shown that polycarbophil and its thiol derivative could extend drug release up to 72 h, while in the case of chitosan and its derivative 12 h release was observed which is less favorable for the purpose described in the study. Moreover, it was found that mechanical properties of the designed systems depended strongly on the type of polymer and also on the content of conjugating agent.

Another frequently applied mucoadhesive polymer is hydroxypropyl methylcellulose (HPMC). Bilensoy et al. [212] formulated hydrogels containing both thermosensitive poloxamer 407 and Carbopol 934 or HPMC as bioadhesive agents. As an active agent, clotrimazole, a poorly water-soluble antifungal agent, was applied. In order to improve the solubility of clotrimazole, an inclusion complex with β-cyclodextrin was used. It was shown that clotrimazole complexation extended its release from hydrogels. Moreover, in the case of Carbopol-based gels, incompatibility resulting fromprecipitation was observed. HPMC-based gels allowed were stable and released the active ingredient in a continuously prolonged manner, which is a promising result in terms of vaginal drug delivery. Aka-Any-Grah et al. [213] presented a study focusing on thermosensitive and mucoadhesive vaginal hydrogels resistant to dilution with vaginal fluids. The investigated formulations contained Pluronic^®^F127 or a mixture of Pluronics^®^F127 and F68 as components providing thermosensitivity. As a mucoadhesive agent, HPMC was used. The results obtained with the use ofan ex vivo animal model indicate that in the case of hydrogels obtained with the combination of Pluronics, mucoadhesive properties were not affected by the dilution. On the other hand, both gels retained their gelling temperature close to 30°C even after dilution.

#### 4.1.3. Thermosensitive Dosage Forms

Thermosensitive gels can be classified in more general terms as stimuli-responsive systems, which are defined as systems undergoing thickening upon physiological conditions. In vaginal drug delivery, the most frequently investigated stimuli-responsive systems remain liquid at room temperature and transform into gels at body temperature. Among the advantages of these systems, easy vaginal administration and suitable contact with folds and crevices of the vaginal mucous membrane should be mentioned. As a result of thermogelation, a more viscous medium is formed, which allows for prolonged release of the active ingredient and also improves the retention time at the administration site. The most commonly applied thermosensitive polymers are poloxamers, with the most frequently used poloxamer 407. It is generally regarded as non-toxic and useful as an excipient in dosage forms designed for the application via different administration routes. It also reveals advantageous thermosensitive properties that enable the formulation of liquid systems transforming into gels at physiological temperature range [155]. As a result, the applied formulation is more resistant to removal mechanisms occurring in the vagina [31]. It is noteworthy that gelation temperature and the properties of gel depend on the composition of the system. An important drawback of poloxamer-based systems in vaginal drug delivery is their poor mucoadhesive characteristics. In order to obtain the required residence time at the administration site, additional bioadhesive excipients are applied. Liu et al. [214] investigated the effect of carrageenan addition to poloxamer 407-based in situ forming a vaginal gel. The aim of the study was to obtain the sustained-release formulation for the delivery of acyclovir, a popular antiviral agent useful in the therapy of genital herpes. Carrageenan was considered a suitable excipient for the vaginal drug delivery system because of its efficacy in the prevention of HIV infections. Rheological studies revealed that the additional macromolecular component did not change the gelation temperature significantly. In vitroacyclovir release experiments showed that the process was slower in the presence of carrageenan, which was related to the retardation of poloxamer 407 dissolution and gel erosion. The observed effect depended on the concentration of carrageenan. The residence time was investigated in vivowith the use of a mouse model. In the performed studies, the carrageenan-enhanced system revealed significantly higher residence time compared to plain poloxamer-based gel.

Rossi et al. [201] performed a study focused on the systems composed of poloxamer 407 and chitosan lactate, as well as chitosan lactate and glycerophosphate mixtures. The investigated systems were loaded with amoxicillin trihydrate. The aim of the project was to obtain thermosensitive vehicles forming gel at physiological temperature for potential application in vaginal mucositis. It was found that the additional macromolecular compounds increased the gelation temperature of poloxamer to the physiological values. Gelation time of poloxamer/chitosan lactate mixture was extended after the dilution with simulated vaginal fluid. In the case of the chitosan derivatives mixture, no such effect was observed. However, the latter one displayed worse elastic properties and better bioadhesion than the poloxamer-based system. Zhou et al. [215] investigated thermosensitive in situ forming hydrogel with baicalein for vaginal administration. The active ingredient was applied as an inclusion complex with hydroxypropyl-γ-cyclodextrin. The hydrogel vehicle was obtained with the use of poloxamer 407, poloxamer 188, sodium alginate, hydroxypropyl methylcellulose (HPMC), and benzalkonium chloride. It was found that the obtained formulation had gelation temperature suitable for in situ vaginal gel and the drug release followed the Peppas equation, which indicates an erosion-based mechanism. The analyzed systems also showedsuitable efficacy in thein vivostudy performed with the use of an animal model. Another study describing poloxamer-based in situ forming a thermosensitive gel with incorporated cyclodextrin complex was presented by DeshkarandPalve [216]. The active ingredient employed in the study was voriconazole, an antifungal agent revealing low solubility in water. In the first step, the inclusion complex of the drug and hydroxypropyl-beta-cyclodextrin was obtained by spray drying. Next, in situ gelling formulation was prepared with the use of poloxamers 407 and 188 and various additional polymers as mucoadhesive agents. In the case of vaginal formulations, the suitable gelation temperature should lie within the 30–35 °C range. It was found that the addition of poloxamer 188 increased gelation temperature, while the addition of mucoadhesive agents had the opposite effect. The most promising properties important in terms of vaginal drug delivery were observed for the formulation containing 0.4% of hypromellose as mucoadhesive polymer. The optimized product revealed a gelation temperature of 31.7 ± 0.1 °C and hadsuitable bioadhesive properties. It was also shown that the application of inclusion complex-based gel instead of a plain one improved voriconazole uptake by tissues, which was shown in in vivo studies. A similar study was performed by Rençber et al. [217] for forming gel loaded with clotrimazole, a popular antifungal agent frequently applied in vaginal candidiasis. The optimized formulation composed of poloxamer 407, poloxamer 188 and hypromellose transformed from liquid to gel at about 34 °C. It was also found that the investigated system revealed suitable mucoadhesive properties and remained at the administration site for 24 h.

### 4.2. Suppositories, Tablets, and Pessaries

Conventional solid vaginal dosage forms, e.g., globules or suppositories, have been in very broad use for many years. Unfortunately, they present a set of drawbacks such as the tendency to irritation and problematic application. Poor retention of the active pharmaceutical ingredients due to the vagina’s self-cleansing or due to leakages may force patients to apply multiple doses daily. All mentioned disadvantages lead to a high inconvenience for the patients and may result in a low adherence and a lack of therapeutic effect. Hence, it has become a challenge for technologists to improve the already existing formulations. The phenomenon of mucoadhesion has become the basis for the development of modern vaginal tablets. Years of research have resulted in the creation of tablets, mini-tablets, pessaries, and other formulations characterized by prolonged vaginal residency, sustained API release, suitable efficacy, and convenience for the patients due to the usage of natural and synthetic polymers with high mucoadhesive properties.

These formulations are mostly used in the topical treatment of bacterial, viral, and fungal infections [129,218,219,220,221,222,223,224], as well as in PrEP (pre-exposure prophylaxis of sexual transmission of HIV) [225,226,227], inflammations [228], atrophic vaginitis [229] and dry vagina [230]. Vaginal tablets have also found application as a carrier of drugs in therapy of cancer [231,232] and probiotics [233]. It is noteworthy that it is crucial for listed therapies to maintain a long vaginal retention time in order to obtain a high effectivity. Hydroxypropyl methylcellulose (HPMC) turned out to be the most commonly used and most promising out of all examined polymers. McConville et al. have proven that the production of tablets containing only one excipient, the sustained-release polymer (HPMC), and releasing efficient doses of tenofovir, an HIV microbiocide, for up to 24 h is plausible [224]. HPMC tablets prepared by Perioli et al. presented prolonged mucoadhesion, very suitable hydration properties with the formation of a homogenous, gelled phase, and, most importantly, prolonged release of benzydamine. Even though Carbopol has brilliant mucoadhesive action, its addition to the formulation eventually resulted in the creation of a spongy, stiff object that could not ensure the linear release of the drug. Other authors supply the data explaining the gains of using HPMC in a combination of polymers [228]. After examining mixtures of HPMC, chitosan, guar gum, and Eudragit RS, Notario-Perez et al. consider joint features of HPMC and chitosan as the most useful. Obtained tablets remained adhered to the vaginal mucosa for 96 h releasing tenofovir for 72 h, which could be used to improve prophylaxis of HIV infection in women from developing countries [225]. A very innovatory solution was proposed by Cevher et al. Combination of mucoadhesive polymers, HPMC or xanthan gum with Carbopol^®^934P prolonged the formulation’s residence time while the usage of cyclodextrin inclusive complexes enhanced itraconazole’s solubility and antifungal activity and also decreased its toxicity, which led to a prolonged drug residency and effectivity [218]. HPMC could also be used in tablets as a mucoadhesive carrier for *Pediococcuspentosaceus SB83,* lyophilized bacteria with antilisterial and pH-reducing activity [233], and spray-dried microspheres packed with clotrimazole used in the antimycotic treatment of the genitourinary tract [219]. In this formulation, the combination of polymers (HPMC with sodium carboxymethylcellulose and Carbopol) helped to avoid burst effects when getting in contact with body fluids and provide controlled release of clotrimazole. Literature finds chitosan very suitable for the formulation of mucoadhesive vaginal tablets. Szymańska et al. investigated the features of clotrimazole tablets prepared with different concentrations of chitosan in three different conditions, using a porcine vaginal mucosa, gelatin disks, and mucin gel. The obtained results confirmed the presence of the bioadhesive properties of chitosan. Preparations containing 25% and 40% of chitosan were regarded as the best candidates for further experiments due to their lengthened residence on the vaginal tissue and stable, prolonged release of clotrimazole [220]. Chitosan may also undergo some behavior-changing modifications. The application of thiolated polymers was proposed by Baloglu et al.who designed preparations with econazole and miconazole nitrates and thiolated poly(acrylic acid)-cysteine [221]. Tablets with synthesized polymer possessed favorable water-uptake ratio and mucoadhesive features. It is worth noticing that thiol groups might be instable toward oxidation. Because of this issue, S-protected chitosan synthesized by thiolating the polymer and protecting the thiol groups from oxidation by an aromatic ligand, was obtained. In this case NAC-6-MNA was attached to chitosan by carbodiimide mediated amide bond. S-protected chitosan presented stronger mucoadhesive properties, prolonged the release of metronidazole, an antiprotozoal and antibacterial agent, from vaginal tablets, and enhanced its antimicrobial activity [129]. Pectins could also be used in modern vaginal drug delivery systems. Baloglu et al. claim that formulation consisting of Carbopol 934 and pectin (2:1) would find a possible application as a carrier for topical acting drugs and as a moisturizer in the dry vagina [230]. This mixture was proven to have the highest mucoadhesive strength and swelling volume. Moreover, it presented the lowest pH reduction. Since the vaginal pH is an issue worth considering, the acid-buffering bioadhesive tablets for vaginal infections must be mentioned. pH rise might be either a symptom or the cause of mixed vaginal infections. Tablets containing sodium monocitrate as a buffering agent ensure a pH 4.4, which is a feature of a healthy vagina. Moreover, their other ingredients, drugs such as metronidazole and clotrimazole and *Lactobacillus acidophilus* spores are efficient in the treatment of genitourinary infections. *L. acidophilus* normally inhabits healthy vagina and it is responsible for its acidic pH. Polycarbophil and sodium carboxymethyl cellulose presented the most suitable behavior for this composition of active agents [222]. The appearance of hyaluronic acid (HA) in the human organism and its biological functions make this polymer a great candidate for vaginal drug delivery. Its structure may be modified to obtain even better properties. Nowak et al. thiolated and preactivated it with 6-mercaptonicotinamide, which effectedhigher stability and enhanced the mucoadhesive values [234]. This solution helps to use the mucoadhesion ensured by disulfide bond formation from thiol groups yet still keeps the substance stable. Moreover, it was proven that HA itself may act as the therapeutic agent, not only the drug carrier. Vaginal tablets containing 5mg of hyaluronic acid were compared with 25 µg estradiol tablets. Both groups turned out to be successful in the therapy of atrophic vaginitis. Even though estradiol is still preferable, HA might find application in patients in which hormonal treatment is contraindicated or undesirable [229].

Mini-tablets have also gained some great attention lately. This dosage form is considered to be an improved formulation of conventional tablets. Numerous polymers were investigated for the production of mini-tablets targeting the vaginal route of administration by Hiorth et al. [231]. The goal of the researchers was to find a perfect formulation to deliver hexyl aminolevulinate hydrochloridum (HAL), a potential topical drug used in the photodynamic therapy of cancer, e.g., cervical cancer, to the vagina. Hydroxypropyl cellulose (HPC) and hydroxypropyl methylcellulose (HPMC) presented adequate mechanical properties, bioadhesive strength, and the release of the drug independent of the vaginal pH, which is crucial because vaginal delivery systems should consider that pH, viscosity, and many other features may vary and are influenced by woman’s age, hormone levels or sexual activity. Methylcellulose, microcrystalline cellulose, and hydroxyethyl cellulose showed no mucoadhesive properties and released whole drug doses within a few minutes. Multiparticulate drug delivery systems such as mini-tablets sized 1–3 mm provide a better spread of the drug inside the vagina, ensure a faster disintegration, longer retention, and even a loss of a few mini-tablets has less impact than in the conventional forms. Mini-tablets might be packed into capsules or applicators, which makes them easy to administer, causes no irritation, and even distribution. These advantages may help to raise the patient’s compliance and maintain a high therapeutic effect.

McConville et al. proposed a simple yet innovative and effective approach to the topic of multipurpose prevention technologies by forming multi-layered tablets helping to avoid pregnancy and sexually transmitted diseases [226]. Thanks to the use of Kollidon^®^ SR and Kollidon^®^ VA, desired drug release profiles and possible in vivo action were obtained. The drugs used in the experiment were antiretroviral dapivirine, contraceptive hormone levonorgestrel, and anti-herpes simplex agent acyclovir. Authors formed tablets consisting of 3–4 layers presenting the immediate or sustained release of API. Results were very promising and indicated a possible introduction of multifunctional multi-layered tablets to the pharmaceutical industry. Prepared tablets, for example, ensured an immediate boost of active substances with a prolonged liberation of dapivirine, which would provide contraceptive action as well as antiviral action. Using these formulations could possibly decrease the number of administered forms and lead to higher patients’ comfort, which would result in higher compliance and satisfying therapeutic effects. Described multifunctional dosage forms could be extremely useful in developing countries to stop the HIV epidemics and immense birth rate.

Osmotic pump tablets (OPTs) are known as an oral drug delivery system, but according to Rastogi et al., the vaginal route of administration is also an option for them [227]. The research team used a potential antiretroviral drug IQP-0528 to form vaginal tablets coated with a bioadhesive polymer (cellulose acetate or cellulose acetate phthalate) with a standard mechanism connected to water intake and release of the drug-loaded gel through an orifice. The results of the study proved that it is possible to prepare tablets delivering active agents in the vagina for 2–5 days. Furthermore, osmotic pump tablets may present the pH-dependent burst release of the drug. This phenomenon can be used to design an HIV-preventing formulation activated by the appearance of semen, which causes a pH change in the vaginal canal. Hence, the OPTs could improve patients’ adherence and effectiveness to PrEP. Even though the pessaries are not so popular, they might be as useful as other vaginal dosage forms. Ceschel et al. developed pessaries of semisynthetic solid triglycerides containing bioadhesive polymers such as polycarbophil, HPMC, and hyaluronic acid sodium salt, which keep the formulations in the vaginal tract for several days without any unwanted reactions. Described pessaries make a suitable carrier for imidazole antimycotic derivatives used to treat frequent mycotic infections, e.g., clotrimazole [223].

### 4.3. Vaginal Rings

There are several rings for contraception available in the pharmaceutical market: NuvaRing^®^ (Merck and Co., Kenilworth, NJ, USA), Progering^®^ (Laboratorios Andrómaco SA, Peñalolén, Chile), Annovera^®^ (TherapeuticsMDInc, Boca Raton, FL, USA), Ornibel^®^/Ginoring^®^ (Exeltis, Madrid, Spain) andEluRyng^TM^(Amneal Pharmaceuticals, Bridgewater, NJ, USA) NuvaRing^®^ is made of ethylene vinyl acetate(EVA) copolymers. Its thickness is about 4 mm, and its diameter is 54 mm. It releases etonogestrel 0.012 mg and ethinyl estradiol 0.015 mgper day. It is used for 3 weeks, and a week interval between the subsequent applications is required. In contrast, Progering^®^ was made of silicone elastomer. It provides 10mg of progesterone per day for 3 months. The subject of some clinical research studies isformulations combining antiviral and contraceptive action. Their aim is to increase the effectiveness of contraception and to prevent sexual transmission of HIV. The purpose of the research conducted by Thurman et al.was to compare the characteristics of vaginal rings and oral contraception [235]. It was found that topical delivery of the drug through the vaginal ring allowed for the reduction in drug doses and was associated with avoidance of a first-pass effect in the liver.

Another advantage of the vaginal rings is less pain and shorter menstruation. They are also comfortable in their stand-alone application and in control. These rings should not be felt during daily activities and do not interfere with sexual behavior. However, they can cause bleeding and vaginal inflammation.

### 4.4. Microspheres

More than 20 years ago, polymeric microspheres were evaluated as a carrier for drugs for vaginal application. Microspheres from hyaluronic acid esters as the carrier for salmon calcitonin were prepared by Rochina and co-workers [236]. They used the solvent extraction method for formulation. Spherical microspheres with smooth surface and diameter of about 10 µm were obtained. The efficiency of incorporation was high; approximately 80% to 90% of the peptide was recovered by extraction from the microspheres. Quantification of the extracted peptide in vivo confirmed that the biological activity of calcitonin was unaffected by the microsphere preparation process. Hyaluronic acid is the polymer that was also used for the study concerned with the possibility to restore the vaginal ecosystem with the microparticles containing probiotics and prebiotics. In 2011 Pliszczak et al. [237] published the results of the study of the design of a new vaginal bioadhesive delivery system based on pectinate-hyaluronic acid microparticles for probiotics and prebiotics encapsulation. Microparticles were produced by the emulsification/gelation method using calcium ions as the cross-linking agent. Inthe beginning, the influence of the main formulation and process parameters on the size distribution of unloaded microparticles wasconducted. Rheological measurements were also performed to investigate the bioadhesive properties of the gels used to obtain the final microparticles. Afterward, an experimental design was performed to determine the operating conditions suitable to obtain bioadhesive microparticles containing probiotics and prebiotics. The encapsulation system could enhance the effects of *Lactobacillus sp*. and protect them during the drying process and storage. The final microparticles had a mean diameter of 137 µm and allowed a complete release of probiotic strains after 16 h in a simulated vaginal fluid at 37 °C. Chitosan-alginate microspheres were developed by Maestrelli et al. [238] for cefixime vaginal administration to overcome problems associated with its oral administration. They prepared microparticles through ionotropic gelation using calcium chloride as the cross-linking agent. Entrapment efficiency increased with drug loading concentration in the starting solution, reaching a plateau at 30 mg/mL indicative of the achievement of an optimal drug-to-polymer ratio. The swelling properties of microspheres increased with the entrapped drug amount, and, interestingly, wateruptake reached its maximum value at the same drug loading concentration of 30 mg/mL. The relationship found between microspheres wateruptake and drug release rate confirmed the microspheres prepared with 30 mg/mL cefixime as the best formulation. Mucoadhesion studies indicated that all formulations assuredin situpermanence longer than 2 h. Microbiological studies showed the relation between cefixime release rate from microspheres and *Escherichia coli* viability reduction. They concluded that evaluated microspheres formulation could be used for effective local treatment of urogenital infections. The other technique for the microsphere preparation is the spray drying process, which was used by different groups and for different polymers [219,239]. Zhang et al. [239] used polymethacrylate salt for the delivery of tenofovir (a model HIV microbicide) and investigated spray-dried mucoadhesive and pH-sensitive microspheres. It has been shown that the sodium or potassium salts of the methacrylic copolymers Eudragit^®^L-100 and S-100 have the potential as a novel low-swellable mucoadhesive material. The optimized formulation has an average size of 4.73 µm with adrug loading of 2% (*w/w*). It has been shown that these microspheres can quickly respond to the pH change, releasing over 90% of the drug within 60 min. The mucoadhesion property of these microspheres is significantly improved compared to the 1% HEC gel formulation. Moreover, the findings in this study reveal that these microspheres arenoncytotoxic and non-immunogenic to vaginal/endocervical epithelial cells. There is also no observable cytotoxic effect on normal vaginal flora. In addition, Gupta et al. [219] used the methacrylic acid copolymers for the microsphere preparation. The aim of the research carried out by their team was the preparation and evaluation of vaginal tablets containing clotrimazole in the form of microspheres. To achieve a long-term therapeutic effect at the site of infection, mucoadhesive polymers: hydroxypropyl cellulose (HPMC), sodium carboxymethyl cellulose, and Carbopol^®^ 934 were used as excipients for the tablets formulation. These microspheres with clotrimazole were prepared by using the spray drying technique with Eudragit RS-100 and Eudragit RL-100. The results indicate that the developed vaginal formulations exhibit controlled drug release. Next to spray-drying, spray-congealing is an interesting method of preparing mucoadhesive microparticles. It was evaluated by Albertini et al. [240], who investigated adhesive microparticles for the vaginal delivery of econazole nitrate. They prepared microparticles based on a lipid-hydrophilic matrix containing both a drug and a mucoadhesive substance with spray-congealing. This method, consisting ofthe atomization of dispersion of the drug in a molten carrier, is a solvent-free technique, which may be advantageous for the preparation of mucoadhesive microparticles. Several mucoadhesive polymers (chitosan, sodium carboxymethyl cellulose, and poloxamers) within the hydrophilic-hydrophobic meltable matrix (Gelucire 53/10, Gattefossé, France) were evaluated. The results showed that the solubility of econazole increased 15 times when it was microencapsulated in the Gelucire 53/10. Albertini et al. concluded that this fact could be correlated to the carrier’s amphiphilic structure (HLB = 10). Once the carrier dissolves in the fluid, it assembles into micelles arranging the hydrophobic part, which includes the drug inside and the hydrophilic portion, which acts as an interface between the simulated vaginal fluid and the drug outside. The addition of poloxamers to the lipophilic carrier provided the same effect, while the addition of chitosan and sodium carboxymethylcellulose to the carrier decreased the API solubility compared to Gelucire 53/10 used alone. This could probably be caused by the fact that the solubilization of the polymers interferes with the effect of the drug wettability and solubility enhancement due to the carrier [240]. In addition, the mucoadhesive properties of the microparticles were investigated. The residence time of the antifungal agent at the infection sites of the vaginal mucosa tissue are very important and mucoadhesive properties can increase it. The particles with the poloxamers showed the best results of the mucoadhesion test. Researchers concluded that spray-congealing technology may be considered as a novel and a solvent-free approach for the production of mucoadhesive microparticles for the vaginal delivery of econazole nitrate.

### 4.5. Pellets

Pellets are a kind of granules. Their sizes range from 300 to 1000 microns. Due to the small size, it can be expected that after vaginal application, they will stay at the surface of vaginal mucosa and will be less susceptible to the force of gravity than vaginal tablets. It is suggested thatpellets can be used as carriers for active substances or as matrices for probiotic bacteria [94,222]. Santos et al. [241] studied whether the addition of the carrier material itself affects the natural protective microflora in the vagina. They used starch-based pellets and lyophilized lactose-based pellets with probiotic bacteria. Gelatin capsules were filled with pellets and prepared for vaginal application. A non-treated control group was included to follow the natural evolution of pH and microflora during the menstrual cycle. No adverse effects on the vaginal and ectocervical mucosa were observed throughout the study period. Researchers concluded that fast-disintegrating starch pellets and lyophilized lactose/skimmed milk are acceptable carrier materials for the vaginal delivery of probiotic strains or other drugs. The research conducted by Poelvoorde et al. [106] focused on the properties of the vaginal non-disintegrating microcrystalline cellulose or disintegrating starch-based pellets. Pellets were administered to patients after being placed in hard gelatin capsules or capsules made from hydroxypropyl methylcellulose (HPMC). Researchers evaluated in vivo behavior (vaginal distribution and retention) and patient acceptability (irritation, discomfort) of pellets. Immediately after application, capsules made of HPMC exhibited better mucoadhesive properties, while the gelatin capsules presented a faster degradation. After the release from the capsules, pellets made from starch decomposed much faster than pellets based on microcrystalline cellulose. Although in vitro disintegration was faster for hard gelatin capsules compared to HPMC capsules, their in vivo behavior was similar as two out of five were still intact 6 h after administration. The authors concluded that slow capsule disintegration would limit the drug release rate. However, the disadvantage could be eliminated if pellets were administered using an applicator with a different design that does not require that pellets be packed in the capsule. Due to the disintegration of starch-based pellets, this formulation probably spread more easily over the vaginal mucosa and was better retained, although high-amylose starch (the main ingredient of the pellets) did not have mucoadhesive properties. A continuation of this work was a comparative analysis of starch pellets and a cream of cetomacrogols conducted by Mehta et al. [104]. They wanted to demonstrate the differences in the deposition of tracer substances after vaginal application. The authors used a technique of gamma scintigraphy and a magnetic resonance imaging method. The studies have been conducted on animals (sheep) and a group of human volunteers. It has been shown that pellets as a result of fast disintegration, covered vaginal epithelium to a degree similar to cream. Further research performed by Metha et al.was based on the preparation of fast-disintegrating tablets with pellets. They hypothesized that tablets would have a shorter disintegration time compared to the capsules, so pellets would distribute faster throughout the vaginal cavity with persistent longer retention [105,242]. The research team evaluated pellets compressed into fast-disintegrating tablets for their distribution and retention in sheep and women using gamma scintigraphy and MRI techniques. In sheep, the tablet disintegration was initiated within 30 min after administration, and within 2–4 h, the entire vagina was covered with the disintegrated pellets with a persistent spread up to 48 h. In women, disintegration was complete within 4 h, and persistent retention was up to 24 h [105]. The suitable intravaginal distribution and long retention time of the disintegrating tablets comprising starch-based pellets provide an interesting vaginal drug delivery platform. These tablets can be further explored as carriers for intravaginal delivery via the incorporation of drugs from different therapeutic groups in the starch-based pellets. On the basis of the research described above, some studies with bioadhesive pellets were prepared by Hiorth et al. [243]. The aim of the study proposed by this group was to develop bioadhesive pellets containing hexyl ester 5-aminolevulinic acid, a precursor of the photoactive substance, with a fast release for vaginal drug delivery. In contrast to the reported disintegrating pellet-based system, the aim of the current study was to develop bioadhesive pellets with a fast release of the active substance. Pellets were produced by extrusion/spheronization, and Carbopol^®^ 934 was used to obtain bioadhesive properties that prolong the residence time in the vaginal tract. Researchers wanted to demonstrate the usefulness of polymers in the treatment of cervical cancer with the photodynamic method. It has been shown that the content of 8% of Carbopol had a positive impact on the mucoadhesive properties of pellets. They showed bioadhesive properties toward vaginal tissue. Investigated pellets were mechanically stable and released the drug load within 20 min in phosphate buffer at pH = 4.0 and 6.8 in the in vitrodissolution test. In addition, the investigated dosage forms exhibited a stability time of 6 to 7 weeks. Proposed delivery systems were suitable for the administration of hexyl ester 5-aminolevulinic acid to the vaginal cavity.

### 4.6. Nanoparticles

In recent years polymeric nanoparticles (NPs) have been widely described as the carriers of drugs for vaginal administration, for locally and systematically acting medicines. NPs are highly stable particles characterized by sizes under 1000 nm [244]. They can encapsulate a wide range of APIs and provide a controlled, prolonged and targeted delivery. The main purposes of the use of NPs in vaginal drug delivery comprise efficient delivery of microbicides, targeted delivery of siRNA, HIVprevention, tumors treatment, delivery of hydrophobic substances, prevention or treatment of sexually transmitted diseases, or delivery of antibiotics [245,246,247,248]. Either natural (e.g., chitosan, hyaluronic acid) or synthetic polymers can be used to prepare such dosage forms. Most researchers focus on synthetic compounds, including poly(lactic-co-glycolic) acid (PLGA),polyethylene glycol (PEG),(meth)acrylate polymers, polyesters (polycaprolactone), and many others. NPs for vaginal administration include a few types of dosage forms: polymeric nanoparticles (NPs), liposomes (LIPs) or cyclodextrins (CD), nanocapsules (NC) nanospheres (NS) [249,250,251,252]. NPs can solve some problems with drug formulation, e.g., poor water/oil solubility, degradation of API, toxicity, or unpleasant organoleptic properties.

#### 4.6.1. Poly(lactic-co-glycolic) Acid

The most described polymer regarding NPs has recently been PLGA. Das Neves et al. tried to formulate the NPs based on PLGA loaded with dapivirine to provide effective delivery of this antiretroviral drug. They used an emulsion-solvent evaporation method and obtained particles with a mean diameter of 170 nm. The formulation was characterized by an initial burst effect up to 4 h followed by a sustained release for 24 h and a lower or at least similar toxicity compared to the free drug. Furthermore, drug retention in cell monolayers was significantly higher for the NPs compared to the free drug [245]. The evolution of vaginally-administered HIVprevention brought the combination of two or more antiretroviral drugs in one medical product. A study conducted by Cuhna-Reis et al. aimed to formulate a novel dosage form containing PLGA-based NPs loaded with tenofovir and efavirenz incorporated into a polymeric film, NPs-in-film. In vivo tests ran on mice revealed enhanced vaginal NPs retention and drug concentrations as well as prolonged drug release, compared to the administration of the same drugs in aqueous solutions. Moreover, the systemic exposure to both drugs was low, and the NPs-in-film was found to be safe for vaginal administration, as it caused no significant genital nor histological changes after application [249]. NPs-in-film has recently gained more interest in the case of antiretroviral delivery for HIV prevention. Work published by Machado et al. describing tenofovir-loaded poly(lactic-co-glycolic acid) (PLGA)/stearylamine(SA) composite NPs with a mean diameter of 127 nm, incorporated into hydroxypropyl methylcellulose/poly(vinyl alcohol)-based film, presented a formulation that could release NPs in just 9 min upon contact with the simulated vaginal fluid (SVF). The release was characterized by an initial burst (around 30% of the drug in 15 min), followed by a sustained release up to 24 h. Described dosage form was found tolerable for vaginal delivery and did not induce any histological changes or pro-inflammatory response, as administered upon 14 days to mice [253]. Many different compounds could be effective for vaginally-administered anti-HIV prevention, not only antiviral APIs. One of those ingredients is siRNA (small interfering RNA). Forthat purpose, Gu et al. tested the poly(ethylene glycol) (PEG)-functionalized poly(D, L-lactic-co-glycolic acid) (PLGA)/polyethylenimine (PEI)/siRNA NPs (siRNA-NP) prepared with a modified emulsion-solvent evaporation method. These PLGA-based NPs were functionalized with HLA-DR antibodies for targeted delivery to dendritic cells and loaded into a biodegradable film. NPs were rapidly released from the film after administration and were able to penetrate the epithelial layer and act locally with targeted cells [246]. PLGA-based NPs are also a suitable carrier for drugs in tumor treatment. Local delivery of chemotherapeutics may reduce systemic adverse reactions. Yang et al. presented a paper describing paclitaxel-loaded PLGA-based NPs (Figure 4). They have made some very interesting and important observation by comparing the mucoadhesion of conventional particles (CP) and mucus penetrating particles (MPP) coated with Pluronic F127. The first ones, as mucoadhesive structures were aggregated in cervical mucus (CVM) and did not achieve a proper distribution close to the tumor cells, while MPPs rapidly penetrated CVM and reached the targeted area. As a result, MPP suppressed the tumor growth more effectively and provided a sustained release, with a minimal effect of the initial burst [247].

Further coating functionalization of the NPs was proposed by Sims et al. They created PLGA-based NPs and coated them with MPG-a cell-penetrating peptide, PEG, MPG/PEG, and vimentin, respectively, and assessed the penetration in 3D human carcinoma cervical cell culture. As a result, they observed an even 66-fold increase in cell internalization in the case of MPG-NPs. The uptake of carcinoma cells was significantly enhanced, and PEG-NPs’ penetration was two-fold higher than MPG-NPs [254].

#### 4.6.2. Polyethylene Glycol

The key polymer for mucus penetration is PEG-neutral, hydrophilic, and minimalizing mucoadhesion, allowing NPs’ compounds to penetrate through the MCV. Such properties of PEG were confirmed by Maisel et al. in their research. They had proven its anti-mucoadhesive feature ex vivo on human CVM and then administered PEG-coated NPs to the cervicovaginal tract of a mouse. As a result, they observed uniform distribution into the vaginal epithelium [255].

Jøraholmen et al. developed PEGylated liposomes with a diameter of 181 nm, able to penetrate MCV, containing interferon alpha-2b for the local therapy of human papillomavirus (HPV). In vitro, they observed no release, but ex vivo tests revealed an elevated TNF α-2b penetration compared to the control group. No mucin interactions were observed, so PEGylated liposomes could reach the deeper epithelium [250]. Another way to treat HPV-induced cervical lesions was proposed by Lechantour et al. They investigated the siRNA-loaded PEGylated lipoplexes for vaginal administration. In vivo studies in mice revealed a complete and sustained coverage of the mucosal epithelium, following a unique vaginal administration of fluorescent PEGylated lipoplexes. Coating lipoplexes with PEG allowed the release of active siRNA into the cytoplasm of HPV-positive cells and consequently induced biological responses and prevented the mucin proteins from aggregation on lipoplexes [256]. PEG is also a suitable polymer for the delivery of photosensitizers. Wang et al. presented boron-dipyrromethene (BDP)-loaded PEG-based NPs, obtained via the solvothermal method. Studies showed that NPs led to an improvement in both cellular uptake and mucus penetration in vitro and in vivo compared to BDP-loaded polymeric micelles. Authors concluded that the excellent photothermal activity of prepared formulation, inducing: tumor apoptosis upon irradiation, its high efficacy, and safety, make the NPs a promising tool in the treatment of severe cervical intraepithelial neoplasia [257].

#### 4.6.3. (Meth)acrylate Polymers

Other compounds suitable for nanocarriers are numerous (meth)acrylate polymers. NPs prepared by Frank et al. were based on Eudragit^®^ RS100 and Eudragit^®^ S100, loaded with the Nile red, as a model of lipophilic substance and incorporated into chitosan gel as an intravaginal medium. NPs sized approximately 200 nm showed a higher penetration of nile red, especially in the case of nanocapsules (Eudragit^®^ RS100), compared to free drugs [172]. The polymer itself is a base in the formulation prepared by Santos et al. The aim of this study was to create a coconut oil-core with the Eudragit^®^RS100 shell NPs, loaded with clotrimazole for candidiasis treatment. NPs were prepared with interfacial deposition of the polymer and presented an average diameter lower than 200 nm and high encapsulation efficiency, close to 100%. In vitro studies revealed higher stability against UV radiation for encapsulated clotrimazole and a prolonged release with no burst effect, whicheffected higher activity against *Candida albicans* and *Candida glabrata* [251]. In the study conducted by Melo et al., a continuation of nano-based antifungal dosage form was presented. The formulation based on Eudragit^®^ RL100 was loaded with amphotericin B (AMB) and coated by hyaluronic acid (HA). NPs obtained by nanoprecipitation and coated by adsorption techniques exhibited approximately 148 nm of diameter and were characterized by a controlled API release during 96 h, followed by the zero-order kinetic profile. This manner provided constant therapeutic doses of AMB that inhibited the in vitro growth of *C. albicans*. In vitro studies ran on rats showed a rapid suppression of the 100% vaginal fungal burden in 24 h. Furthermore, it was suggested that HA coating interacted with the CD44 receptor on epithelial cells, and as a result, NPs might be internalized to efficiently exert their antifungal action [248].

#### 4.6.4. Polyesters (Polycaprolactone)

Polycaprolactone (PCL) is a base polymer of NPs prepared in the work of Frank et al. Authors aimed to increase the adhesion and penetration through the vaginal mucosa in order to treat HPV infections bycombining nanocarriers and mucoadhesive semisolids. For this purpose, two formulations were obtained: (a) chitosan-coated poly(e-caprolactone)-nanocapsules incorporated into hydroxyethylcellulose gel, and (b) poly(e-caprolactone)-nanocapsules incorporated into chitosan hydrogel. As a result of their studies, chitosan-coated NCs combined with mucoadhesive gel wasdescribed as the most promising dosage form considering the permeability, mucoadhesion, and drug retention [258]. Different methods were used to obtain a similar goal by Varan et al. in their paper. Both anticancer (paclitaxel) and antiviral (cidofovir) drug combination was manufactured by inkjet printing onto an adhesive film to locally treat cervical cancers induced by HPV infections. Poorly soluble paclitaxel was encapsulated into a cyclodextrin complex, and cidofovir was encapsulated in polycaprolactone NPs sized below 200 nm (Figure 5). In vitrostudies showed effectiveness on human cervical adenocarcinoma cells with the synergistic effect of two drugs. Other advantages of such dosage form are personal dose fixing and prolonged-release possibilities [252].

PCL is also suitable to deliver the antifungal itraconazole. The authors obtained NCs and NSs by the nanoprecipitation method. Formulations were characterized by a high encapsulation efficiency, 99% and 97%, respectively, and the size below 190 nm and 120 nm, respectively. Then, NPs were incorporated into a vaginal viscous formulation and administered to female mice infected by *C. albicans.* Results showed high efficiency in fungal reduction only for NCs, compared to NSs and the drug solution. Furthermore, histological analysis showed significant differences between tissues correlated with inflammatory cytokines levels. NC-treated animals showed reduced cytokine levels, while NS- and solution-treated mice showed increased levels of cytokines and tissue inflammation. This shows a high potential of PCL-NCs in itraconazole treatment improvement and cytotoxicity reduction [259].

#### 4.6.5. Polymers of a Natural Origin

Polymers of a natural origin are also investigated as a base of nanocarriers. One of the most described natural polymers is chitosan. Jøraholmen et al. prepared chitosan-coated LIPs loaded with clotrimazole for the local vaginal infection treatment in pregnant women in order to avoid systemic absorption. Sonicated liposomes were coated with chitosan in three different concentrations: 0.1%, 0.3%, and 0.6%. Their sizes ranged from 100 to 200 nm. In vitro release study showed a prolonged release of clotrimazole while the ex vivo experiments performed on the pregnant sheep vaginal tissue revealed that coating with chitosan assured an increased API tissue retention and a reduced penetration compared to the control. Lower chitosan concentration provided a higher mucoadhesive potential [250]. Chitosan NPs also have the potential to deliver peptides. In the work of Marciello et al., such NPs obtained by the ionotropic gelation method with pentasodium tripolyphosphate were incorporated into freeze-dried cylinders. In vitro release studies of insulin as a model peptide showed a rapid release with the burst effect, where more than 50% of the peptide was released during the first 15 min of the experiment. Nearly 100% of the peptide was released within 30 min. The nanoparticles’ ability to promote the peptide penetration into the vaginal mucosa was also proven during the studies (Figure 6) [260].

The same method to obtain chitosan NPs was used by Rossi et al. They aimed to develop a formulation containing amoxicillin trihydrate-loaded NPs and to incorporate it into a fast-dissolving matrix for the treatment of atrophic vaginitis. NPs showed an in vitro mucoadhesion, wound healing (due to the presence of ascorbic acid), and improved antimicrobial properties, as referred to as a solution. Furthermore, NPs released in SVF from the matrix presented an unchanged size [261].

### 4.7. Vaginal Films

Vaginal filmsare another form of drug delivery systems formulated using a variety of polymers for achieving mucoadhesion as well as desirable release profiles of the actives. As a result, they are capable of combating some of the challenges presented in vaginal drug delivery: pH, cervical secretions, permeability, etc.

The films are solid dosage forms mostly prepared using aqueous polymers and plasticizers and may containan active ingredient [12]. They are usually preferred over traditional semisolid formulations because of patient-friendly application, better residence time, higher stability over a variety of conditions, and even aesthetic advantages [194,262,263]. They are usually soft, flexible, preferably colorless and odorless, and disperse or dissolve upon coming in contact with vaginal fluids to adhere and retain in the vaginal mucosa for prolonged durations [264]. Several polymers have been investigated over the years for formulating vaginal films, the most commonly used ones being polyvinyl alcohol [265,266,267,268,269,270,271] and cellulose derivatives [272,273,274] or a combination of both [194,249,253,275,276,277,278]. Other polymers have also been explored with or without the previously mentioned common options [223,263,279,280]. Cellulose acetate phthalate was initially considered an inactive pharmaceutical ingredient with film-forming properties but was later found to have antiviral activity of its own against genital herpes virus [281] and HIV [282] and has been formulated into a composite microbicidal vaginal film with hydroxypropyl cellulose while maintaining activity [272]. The most commonly used plasticizers to achieve desirable film characteristics are polyethylene glycol (PEG) and glycerine, but others have been used as well.

The most widely employed method for the fabrication of vaginal films is the solvent casting method [264], where solutions of polymers, plasticizers, actives, and other excipients are poured in a mold that is cast into films upon drying either at room temperature or under mild heating. The films are then cut into individual films if not poured directly into single film-sized molds [264].

More recently, Regev et al. developed microbicidal vaginal films containing dapivirine using the hot-melt extrusion (HME) technique while using a quality-by-design (QbD) approach [283]. When compared with solvent cast (SC) films of similar compositions, it was found that the HME films were thicker, heavier, had lower water content, and disintegrated faster than SC films while still being within acceptable attribute ranges for vaginal films. Even if not significantly advantageous over solvent casting, HME could be helpful to achieve continuous manufacturing and easier scalability.

The traditional vaginal film formulations are often modified to achieve additional advantageous properties. Cautela et al. formulated three types of composite films for delivery of two anti-HIV drugs: a single-layer film, a double-layer film with two halves of the film cast separately and bound together using a hydraulic press and nanoparticles (NPs) in film that contained a layer of drug-loaded NPs evenly dispersed and sandwiched between two non-drug films by means of a hydraulic press. The fastest drug release followed by an initial burst release was seen with the single-layer films, followed by the double-layer film, and finally, the slowest release was observed with the NPs-in-film, suggesting different ways to achieve a variety of drug release profiles as per therapeutic need [268]. Another recent innovationhas been demonstrated by Machado et al.in the form of a vaginal sheet, which consisted of a gelatin-based gel freeze-dried to form a sheet [284]. Formulated with lactose as a model powdered ingredient, the vaginal sheet was able to hold desired organoleptic characteristics, including texture, and showed buffering capacity with vaginal fluid stimulant (VFS). Garg et al.described films as three basic types: fast-disintegrating (flash release or flash dispersal films), non-disintegrating films (can be used to control residence time when combined with fast-disintegrating films), and medium disintegrating films [285]. Notario-Pérez et al. classified films based on film solubility: hydrophilic films (containing hydrophilic polymers and polar plasticizers) and non-water-soluble films (containing hydrophobic or amphiphilic polymers (to prolong drug release and residence time); based on film structure, monolayer single-polymer films, monolayer blend films, bilayer films, and multilayer films; and based on release mechanisms, stimuli-responsive or smart films [286].

Garg et al. suggested that a film formulation should be used as the delivery system when an intermediate onset of action is required in a product with moderate stability (in specialized packaging) when stored in ambient conditions [287]. Due to maximum efficacy, some crucial factors must be considered during the development of vaginal films, e.g., mucoadhesion, disintegration, drug release, stability, physicochemical characterization, mechanical properties.

Vaginal films have been investigated for a number of applications, with the most common ones being microbicidal owing to the most frequent diseases that affect the vagina. A strategy for employing microbicidal vaginal films is as a prophylactic tool for sexually transmitted diseases (STDs) [264,286]. Films have gained acceptability and preference over the traditional vaginal formulations for their patient-friendly use [288]. A variety of active ingredients have been loaded in vaginal films to find antiviral [281,289], antibacterial [12,290], and antifungal [291,292] applications. Vaginal films have also found their use as contraceptive devices due to their discrete nature and their ability to not only form physical barriers but also that of carrying spermicides. Frankmanet et al. showed the clinical relevance of C-film, a nonylphenoxypolyethoxyethanol vaginal film that was a reliable alternative to traditional contraceptive devices at the time [293]. Garg et al. developed and characterized sodium polystyrene sulfonate (PSS) noncytotoxic contraceptive films [194]. Multipurpose prevention technologies (MPT) that work as contraceptives while providing protection against STDs have gained a lot of interest fromresearchers as well as consumers [294,295]. Nonoxyl-9 (N-9) was one of the earliest marketed agents that showed multipurpose applications but did not prove to be very safe in later studies but has led to the development of other products with better safety and efficacy [296]. A combination of drugs hasbeen used in film formulations for MPT benefits or to have higher efficacy compared to single drug delivery systems [249,266,277,285]. Monoclonal antibodies (mABs), whichhave been shown to be effective in the prevention of infections after vaginal application, have found their way into a vaginal multipurpose prevention filmMB66 [270]. The film developed by Politch et al. is a combination of two human mABs, VRC01, and HSV8, together providing protection against HIV-1 and HSV-1 and 2, and hasshown safety and efficacy in a Phase I clinical trial [270]. Another explored the application of vaginal films is for the treatment of female sexual arousal disorder (FSAD) as shown by Yoo et al.for mucosal delivery of nitric oxide (NO) [279]. Their s-nitroglutathione (GSNO) films displayed significantly enhanced duration of action of GSNO, which is an NO donor, and also showed enhanced vaginal blood perfusion in a rat model without any cytotoxicity. Patents have been filed on vaginal films for a variety of applications, such as drug delivery of analgesics, anesthetics, anti-inflammatory, antimicrobials, vitamins, hormones, proteins, etc. [297]; deodorizing [298]; contraception, infections, itch relief, cleansing, and lubrication [299]; *Lactobacillus* for prevention of infections or restoration of flora [300]; pH regulation, drug and hormone delivery [301] and more. A notable mention is the application of the mixed solvency concept by Gahlot&Maheshwari for the preparation of metronidazole vaginal films for bacterial vaginosis (BV) [274]. They were able to solubilize the poorly water-soluble drug using a combination of aqueous solubilizers, niacinamide: sodium benzoate: urea: caffeine (15:10:10:5), which according to mixed solvency concept exhibit a synergistic solvent action in combination while also minimizing the amount of each individual excipient in the formulation [274]. In recent advancement of vaginal film formulations, Notario-Pérez et al. developed “smart microbicide” films that are pH-sensitive and could change the release behavior of the drug to a rapid release in vaginal pH conditions after ejaculation (pH 7–8) from a sustained-release profile under typical vaginal conditions (pH 4–5) [302].

## 5. Conclusions and Future Directions

An impressive evolution of novel manufacturing techniques thatprovide entirely new possibilities to develop very sophisticated formulations and platforms fordrug delivery via the vaginal route, followed by local action or systemic effects, can be recently observed (Table 1).

Advantageous features of the vagina as a potential drug administration site have been recognized and used for many years [29], even though the full potential of the vaginal route seems to be underestimated, as it was noted in other comprehensive reviews related to this subject [262]. The most important research area in vaginal drug delivery focuses on local action, which is a natural consequence of numerous diseases and conditions, including bacterial, viral, and fungal infections or vaginal atrophy and dryness, potentially occurring in vagina. However, vaginal drug delivery systems are also applied to exert systemic actions. The most extensively investigated directions are hormone delivery in menopause management and contraception. Moreover, vaginal formulations are also investigated as potentially useful preventive agents in sexually transmitted diseases including HIV infections prevention. However, some of the vaginal physiological features are challenging in terms of drug delivery, especially the high variability of vaginal conditions, such as amount, composition and pH of vaginal fluids. On the other hand, these conditions also vary intraindividually, which may also contribute to different therapeutic responses in the same subject.

As far as the polymers applied in vaginal drug delivery are concerned, the most important research directions are related to the increase in the residence time in the vaginal cavity. For this purpose, mucoadhesive and smart polymers, increasing viscosity upon contact with the physiological environment, are frequently investigated. It is important to note that the polymer properties can be tailored with the use of chemical modifications to obtain the desired product characteristics [303]. However, it must be emphasized that in the case of any newly synthesized excipient, detailed studies regarding potential toxicity and irritancy toward vaginal mucosa are necessary. Most of the available literature reports focus on formulation studies and investigate the properties of the analyzed systems with the use of in vitro tests and ex vivo animal models, which is not sufficient for the proper safety evaluation. Therefore, in future product evaluations, more attention should be paid to the toxicity of the potential vaginal drug delivery systems. Moreover, there are also other safety concerns regarding vaginal products. In the case of hormone-loaded systems, the risk of active ingredient transfer to a sexual partner during coitus has been indicated [304]. Another important issue is unwanted systemic exposure to the active ingredient as a result of the drug absorption from topical intravaginal products to the systemic circulation. However, even though this phenomenon was indicated elsewhere as a potential therapeutic problem, according to the available literature reports, only a small amount of the active ingredient can be absorbed and exert the systemic effects and the risk of serious side effects is negligible [305].

It must also be emphasized that the research studies cited in this paper do not present uniform testing protocols for the drug release and permeation across the vaginal mucosa. Some of the investigated formulations are tested for drug release only, and in some cases, excised animal mucosa is applied in ex vivo permeation tests. However, the exact correlations between the ex vivo and in vivo animal studies and pharmacokinetic parameters in humans are scarce, and this research area requires more scientific attention in future studies. As it was already mentioned, animal models usually display some differences with respect to human anatomy and physiology, which indicates the need to develop novel biosimilar models for drug permeation testing.

A large number of concepts and ongoing projects, as reflected in the numerous scientific papers, is consequently pushing the technology of polymer-based vaginal formulations to the new level of coping with such inconveniences as poor bioadhesion, short residence time, thinning with vaginal fluids, insufficient dispersion in the vaginal cavity or rapid drug release. However, the proper safety testing protocols included at the initial stage of the project related to vaginal drug delivery would be useful in further pharmaceutical product development.

**Table 1 pharmaceutics-13-00884-t001:** Polymer-based vaginal formulations.

API(s)	Formulation	Polymer(s)	References
---	bioadhesive tablets	Carbopol^®^934, pectin, PVP, ethyl anhydrated maleic resins	Baloğlu et al. (2003) [230]
---	dendrimers	SPL7013 BHA.lys15lys16(NHCOCH2O)1-(3,6-naphth(SO3Na)32 (BHA: benzhydrylamine)	Gong et al. (2005) [306]
---	gel	Pluronics^®^ F127 and F68, HPMC chitosan-thioglycolic acid conjugates; polycarbophil, Pluronic F 127	Aka-Any-Grah et al. (2010) [213] Friedl et al. (2013) [307] Podaralla et al. (2014) [308]
---	gel-microemulsions	carageenan, xanthan gum	D’Cruz et al. (2001) [309]
---	microparticles	CMC	Kejdušová et al. (2015) [310]
---	mucoadhesive sponges	HEC 250M	Furst et al. (2015) [77]
---	nanoparticles	chitosan, poly(isobutylcyanoacrylate);	Pradines et al. (2015) [311]
---	peptide-derivatized dendrimers	---	Luganini (2011) [312]
---	tablets	hyaluronic acid	Ekin et al. (2011) [229]
4′-ethynyl-2-fluoro-2′-deoxyadenosine (EFdA)	film	PVA; HPMC E5	Zhang et al. (2013) [275]
abacavir	bioadhesive film	Alg-Na, HPMC	Ghosal et al. (2014) [289]
	film	Alg-Na; HPMC E5; HPMC-PVP blend	Ghosal et al. (2014) [289]
acyclovir	in situ gel	poloxamer, carageenan, Carbopol 934p-NF	Liu et al. (2009) [214]
	insitu forming hydrogel	hyaluronic acid, poloxamer F127 F68	Mayol et al. (2008) [166]
acyclovir, ciprofloxacin	gel	chitosan citrate	Bonferoni et al. (2008) [208]
amoxicilin	hydrogel	PEG-dendrimercrosslinks	Navath et al. (2011) [313]
	fast-dissolving matrix	PVP	Rossi et al. (2017) [261]
amphotericin B	insitu gel	poloxamer 407, HPCD	Kim et al. (2010) [314]
amphotericin, fluconazole	liquid crystal precursor mucoadhesive system	chitosan, poloxamer	Salmazi et al. (2015) [315]
arctigenin	liposome-based gel	pH-sensitive liposomes	Chen et al. (2012) [316]
baicalein	insitu gel	poloxamer, HPCD	Zhou et al. (2013) [215]
benzydamine HCl	tablets	HPMC, Carbopol 940	Perioli et al. (2011) [228]
camptothecin	nanoparticles	PLGA	Blum et al. (2011) [317]
chlorhexidine	inserts	chitosan, CMC	Bigucci et al. (2015) [318]
chlorhexidine digluconate	freeze-dried polimer complexes	Alg-Na, chitosan	Abruzzo et al. (2013) [319]
cisplatin	nanofibersgels, films	PLA, PEO, HPMC, Carbopol	Zong et al. (2015) [320]
clindamycin phosphate	bioadhesive system	HPC, xanthan gum	Dobaria and Mashru (2010) [321]
clomiphenecitrate	gel	polycarbophil-cysteinę and chitosan-thioglycolic acid conjugates	Cevher et al. (2008) [211]
clotrimazole	gel	Pluronic^®^F127, polycarbophil, Carbopol. HPC, PVP poloxamers 407 and 188	Bilensoy et al. (2006) [212] Chang et al. (2002) [322]
	film	HPC, Alg-Na	Mishra et al. (2016) [323]
	nanocapsules	Eudragit RS100	Santos et al. (2014) [251]
	tablets	chitosan, (silicified MCC, potato starch,	Szymańska et al. (2014) [220]
	tablets with microspheres	Eudragit RS-100 and RL-100	Gupta et al. (2013) [219]
clotrimazole, metronidazole	acid-buferring tablet	polycarbophil, HMPC	Alam et al. (2007) [222]
coumarin-6	nanoparticles	PLGA	Cu et al. (2011) [324]
CSIC	film	PVA-HPMC K4M blend; PEG 4000	Gong et al. (2017) [276]
dapivirine	film	PVA, HPMC 4000, PEG 8000 PEO, HPC	Akil et al. (2011) [278] Regev et al. (2019) [283]
	nanoparticles	poly(ε-caprolactone) PLGA	Neves et al. (2014) [325] Neves and Sarmento (2015) [245]
dapivirine and tenofovir	film	PVA	Akil et al. (2014) [266]
disulfiram	tablets	MCC, maize starch	Baffoe et al. (2014) [232]
doxorubicin	nanoparticles	carboxyl modified polystyrene	Ensign et al. (2013) [326]
econazole	film	gelatin, PVP, Soluplus^®,^ and Gelucire^®^ evaluated for solid dispersions	Dolci et al. (2020) [292]
	microparticle-loaded gel	chitosan lactate, poloxamer 407, Eudragit RS	Parodi et al. (2013) [327]
econazole and miconazole nitrate	gel	chitosan	Şenyigit et al. (2014) [209]
econazole nitrate, miconazole nitrate	tablets	thiolated poly(acrylic acid)-cysteine (PAA-Cys) conjugate	Baloglu et al. (2011) [221]
econazole nitrate	microparticles	chitosan, Na-CMC, poloxamers	Albertini et al. (2009) [240]
Efda and 5-chloro-3-phenylsulfonylindole-2-carboxamide (CSIC)	film	PVA, HPMC E5, PEG 4000	Zhang et al. (2015) [277]
fluconazole	film	HPMC chitosan:pectin (75:25)	Kumar et al. (2013) [273] Mishra et al. (2017) [291]
fluorescent labeled NPs	film	PVA, carrageenan, PEG	Traore et al. (2018) [267]
FSAD S-nitrosoglutathione (GSNO)	film	Carbopol 934P, HPMC, PEG	Yoo et al. (2009) [279]
griffithsin/carrageenan	fast-dissolving insert	carrageenan, HEC, xanthan gum	Lal et al. (2018) [113]
griffithsin/carrageenan	fast-dissolving insert	carrageenan	Derby et al. (2018) [328]
hexylaminolevulinate hydrochloridum	pellets	MCC, Carbopol	Hiorth et al. (2012) [243]
	bioadhesive mini-tablets	MC, HEC, HPC, MCC	Hiorth et al. (2014) [231]
HIV; IQP-0528	film	PLGA:Eudragit S 100 nanoparticle encapsulated drug in polymeric films	Srinivasan et al. (2016) [329]
HIV and VC; Ebselen	rapidly soluble film	β-cyclodextrin, PVA, Soluplus^®^	Vartak et al. (2020) [269]
HIV-1 reverse transcriptase inhibitors UC781, tenofovir	gel	HEC, Carbopol^®^974P	Mahalingam et al. (2010) [330]
IQP-0528 (non-nucleoside reverse transcriptase inhibitor)	osmotic pump tablets	HPC, CAP, Carbopol 974P	Rastogi et al. (2013) [227]
itraconazole	bioadhesive tablets	cyclodextrins	Cevher et al. (2014) [218]
	film	HPC, PEG 400	Dobaria et al. (2009) [331]
	insitu gel	HPMC E50, poloxamers 188 and 407	Karavana et al. (2012) [332]
itraconazole, tea tree oil	thermosensitive gel	Lutrol^®^F127	Mirza et al. (2013) [333]
lactic acid	gel	chitosan poloxamer 408, chitosan	Bonferoni et al. (2006) [207] Rossi et al. (2014) [201]
	tablets	MC, chitosan	Małolepsza-Jarmołowska (2007) [334]
M48U1 anti-HIV microbicide	gel	Pluronic^®^F127, F68, HPMC	Bouchemal et al. (2013) [335]
maraviroc and emtricitabine	non-aqueous gels	silicone elastomer	Forbes et al. (2014) [336]
metronidazole	film	HPMC E5 S-protected gellan gum	Gahlot and Maheshwari (2018) [274] Jalil et al. (2019) [337]
	gel	chitosan, HEC, 5-methylpyrrolidinone-chitosan (MPCS); PF-127	Perioli et al. (2008) [338] Ibrahim et al. (2012) [339]
	tablets	chitosan, Alg-Na, MCC, CMC; chitosan (FG90C), polyvinylpyrrolidone (PVPK90) and polycarbophil (PCPAA1)	El-Kamel et al. (2002) [340] Perioli et al. (2009) [341]
	tablets with preliposomes	MCC, starch, pectin, chitosan	Vanić et al. (2014) [342]
microbicidal-STD pathogens (HIV, HSC); bacteria associated with BV Cellulose acetate phthalate (CAP)	film	HPC	Neurath et al. (2003) [272]
microbicidal for HIV and HSV; mAB VRC01-N; mAB HSV8-N	film	PVA, maltitol, polysorbate 20	Politch et al. (2021) [270]
icrobicides PHI-113, PHI-346, PHI-443	self-emulsyfying gel	PEG 400, MCC, xanthan gum	D’Cruz et al. (2005) [343]
MIV-150/zinc acetate/carrageenan	gel	carrageenan	Friedland et al. (2016) [344]
MIV-150/zinc acetate/carrageenan	gel	carrageenan	Kenney et al. (2012) [345]
maraviroc	electrospun fibers	PVP, PEO	Ball andWoodrow (2014) [346]
Na fluorescein, nile red	nanoparticles	Eudragit S-100, PVP	Yoo et al. (2011) [167]
natamycin	tablets	HPMC, xanthan gum, Carbopol 934 P, cyclodextrins	Cevher et al. (2008) [347]
nile red	polymeric nanocapsules in hydrogel	chitosan, Eudragit	Frank et al. (2014) [172]
nystatin	gel	poly(acrylic acid)-cysteine conjugate and the new poly(acrylic acid)-cysteamineconjugate	Hombach et al. (2009) [348]
	microparticles	Alg-Na, poloxamer 407, chitosan	Martín-Villena et al. (2013) [349]
ovoalbumin	microparticles	PLGA	Kuo-Haller et al. (2010) [350]
	gel	chitosan, HPMC K100M, Pluronic F 127	Tuğcu-Demiroz et al. (2013) [210]
polyherbal microbicides	cream	Alg-Na, xanthan gum	Talwar et al. (2008|) [351]
polystyrene sulfonate (PSS)	film	HPMC, HEC, PVA	Garg et al. (2005) [194]
probiotic microorganisms	microparticles	pectinate, hyaluronic acid	Pliszczak et al. (2011) [237]
progesterone	hydrogel	glycolchitin	Almomen et al. (2015) [352]
	mucoadhesive emulsion	cyclomethicone pentamer	Campaña-Seoane (2014) [353]
propranolol HCl	gel	guar gum, Alg-Na, xanthan gum, HPMC 4000, Na-CMC, carbomer 934, 940	Tasdighi et al. (2012) [354]
proteins, insulin	flux controlled pump, pellets	HEC, HPC, CG,	Teller et al. (2014) [355]
pyrimidinedione IQP-0528	film	PVA	Ham et al. (2012) [271]
raltegravir + efavirenz	nanoparticles loaded gel	Pluronic^®^F127 and F68	Date et al. (2012) [356]
saquinavir	nanoparticles loaded gel	HEC, PLGA, PVA	Yang et al. (2013) [357]
sertaconazole	microemulsion-based gel	Carbopol 940	PatelandPatel (2012) [358]
sertaconazole	tablets	Cbp 934P, CH, CMC-Na, Alg-Na, MC, HPMC, HPC	Patel et al. (2011) [359]
siRNA-loaded nanoparticles with anti-HLA-DR antibody (siRNA-NP-Ab)	film	PLGA-PEG/PEI/siRNA-NP PVA-λ-carageenan film	Gu et al. (2015) [246]
SPL7013 sulphonated dendrimer	gel	Carbopol^®^	Mumper et al. (2009) [360]
STDs sodium dodecyl sulfate (SDS)	film	Carbopol 934P, HPMC, PEG	Yoo et al. (2006) [263]
*Streptococcus* vaccine	microparticles	Resomer, RG 503 PLG	Hunter et al. (2001) [361]
tebuconazole	nanoparticles	tetraethylorthosilicate	Mas et al. (2014) [362]
tenofovir	film	drug-loaded PLGA/SA composite NPs incorporated into a PVA-HPMC film; EC: xanthan gum (2:1)	Machado et al. (2016) [253] Cazorla-Luna et al. (2020) [363]
	microparticles	Eudragit S-100 sodium salt	Zhang et al. (2013) [239]
	nanoparticles	chitosan; hyaluronic acid	Meng et al. (2011); Meng et al. (2014) [364,365] Agrahari et al. (2014) [366]
	tablets	HPMC, Kollidon SR	McConville et al. (2013) [224]
	SLN	PAA	Alukda et al. (2011) [367]
	films	HPMC-Zein (1:5) blend, PEG Eudragit RL, RS, L and S	Notario-Perez et al. (2019) [280] Notario-Perez et al. (2021) [301]
tenofovir + efavirenz	film	drug-loaded PLGA NPs in HPMC-PVA films	Cunha-Reis et al. (2016) [249]
tenofovir disoproxil fumarate and emricitabine	film	Eudragit^®^L100 NPs in PVA films	Cautela et a. (2019) [268]
tenofovir, emtricitabine	tablets	microcrystalline cellulose, crospovidone, hydroxyethyl cellulose	Clark et al. (2014) [368]
tenofovir, maraviroc	dendrimers	carbosilane	Sepúlveda-Crespo et al. (2014) [369]
tenofovir, maraviroc, dapivirine	film	sodium CMC, HPMC, HEC; PVA, PVP-K90, PVP-K30	Akil et al. (2015) [370]
tenofovir, tenofovirdisoproxil fumarate	nanoparticles	PLGA, Eudragit	Zhang et al. (2011) [371]
tioconazole	film	chitosan-HPMC, PEG 400	Calvo et al. (2019) [372]
UAMC01398	solid dispersion film	HPMC, PEG 400	Grammen et al. (2014) [373]

## Figures and Tables

**Figure 1 pharmaceutics-13-00884-f001:**
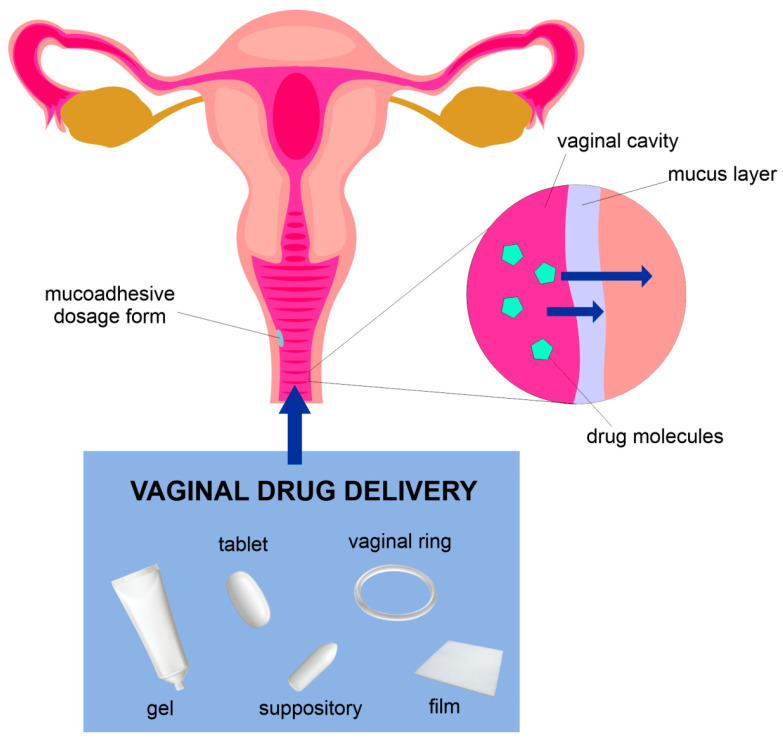
Vaginal drug delivery route.

**Figure 2 pharmaceutics-13-00884-f002:**
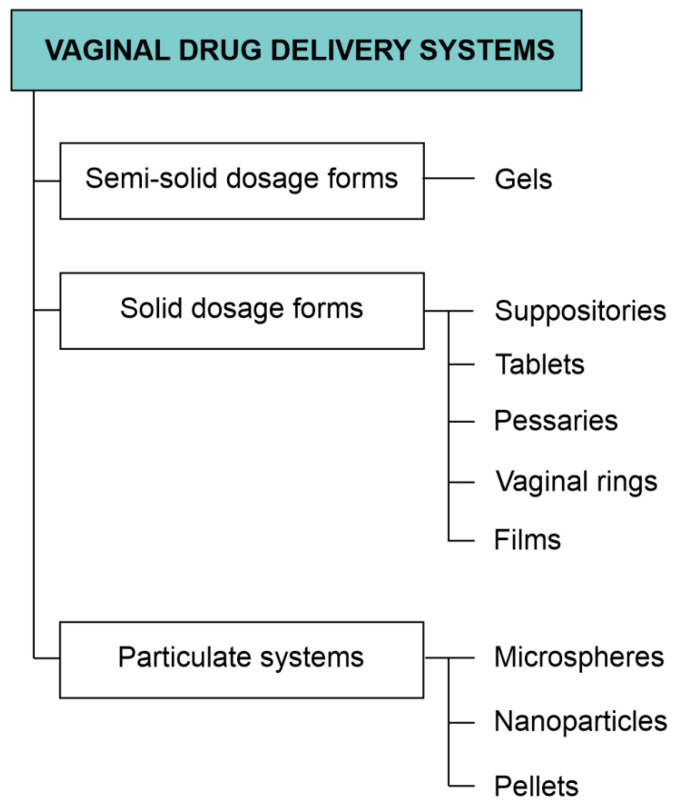
Classification of the most extensively investigated vaginal dosage forms.

**Figure 3 pharmaceutics-13-00884-f003:**
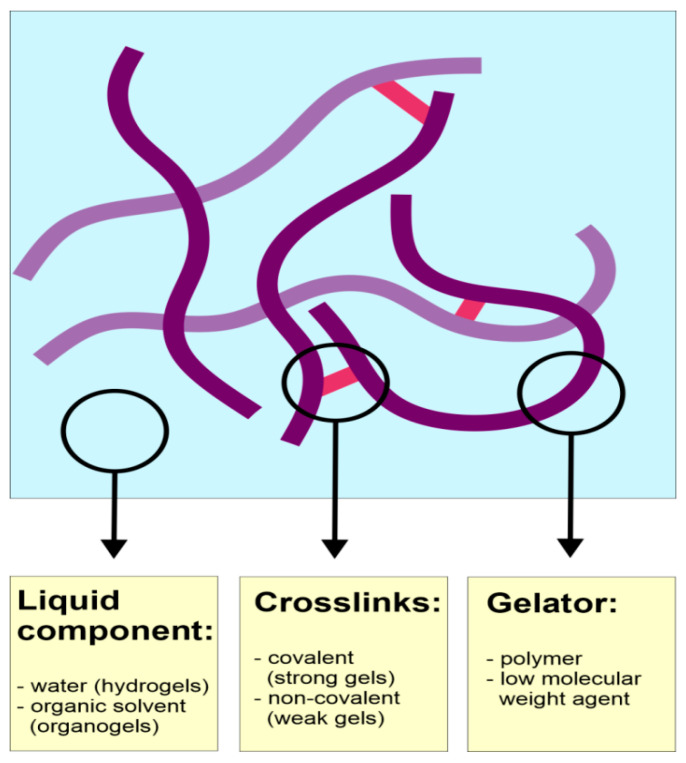
Classification of gels according to different criteria.

**Figure 4 pharmaceutics-13-00884-f004:**
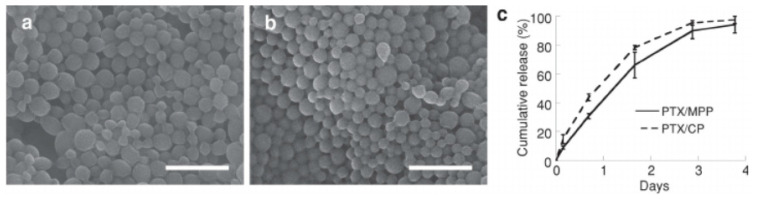
Characterization of PTX/PLGA nanoparticles in vitro. SEM images of (**a**) PTX/MPP and (**b**) PTX/CP; scale bar represents 1μm. (**c**) Cumulative in vitro release of PTX from PTX/PLGA nanoparticles over time [247].

**Figure 5 pharmaceutics-13-00884-f005:**
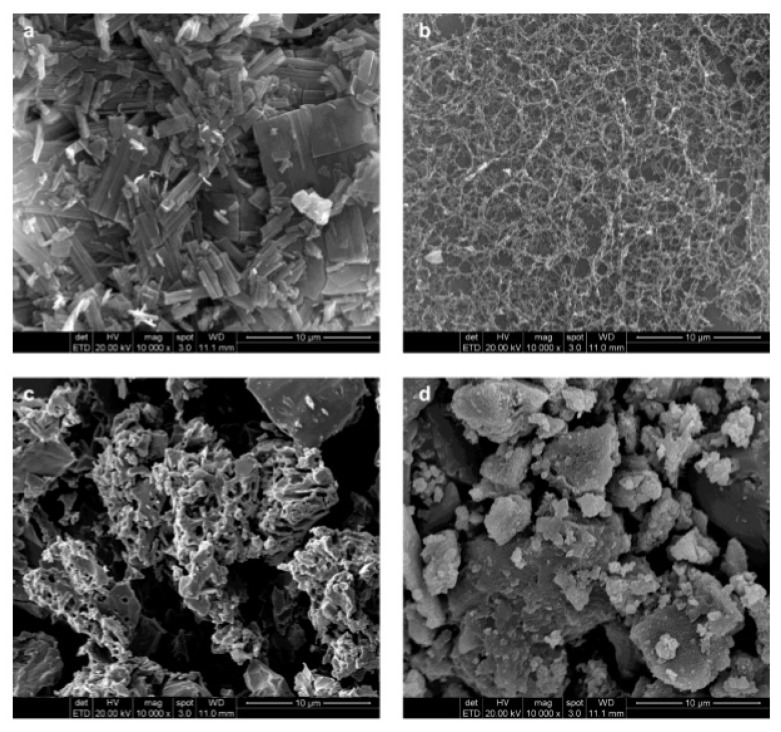
SEM photomicrographs of (**a**) free PCX (**b**) PCX:HP-β-CD, (**c**) PCX:6-O-CaproβCD, and (**d**) PCX:PCβCDC6 inclusion complexes [252].

**Figure 6 pharmaceutics-13-00884-f006:**
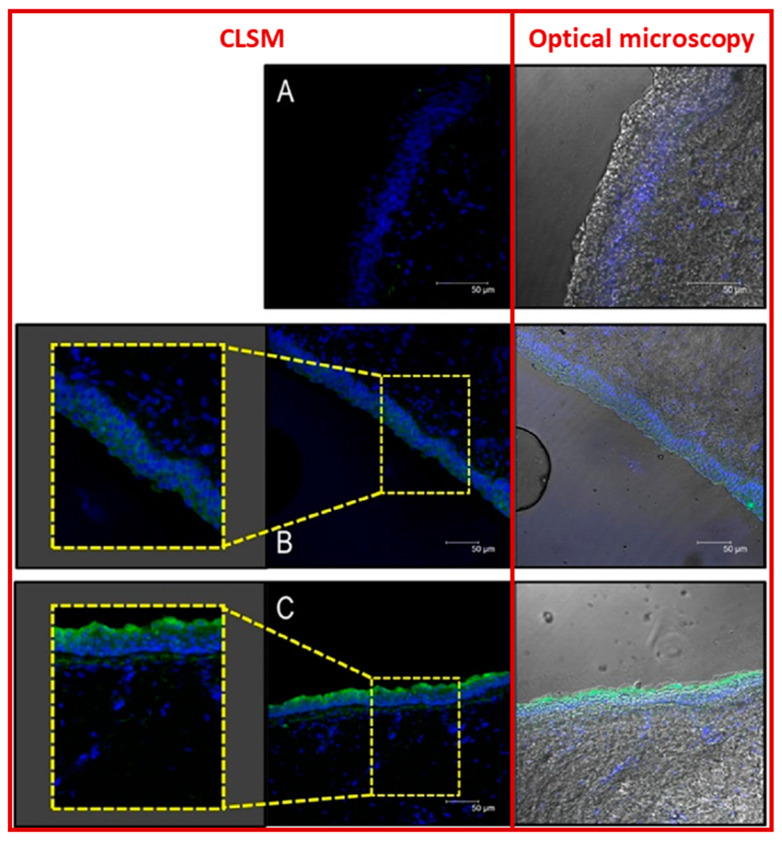
CLSM (confocal laser scanning microscopy) images of porcine vaginal mucosa alone (**A**), placed in contact with a solution of insulin-FITC (**B**) and with a suspension of insulin-FITC loaded CS NPs (**C**). Zoom of a CLSM picture detail and an overlap with a micrograph of the same mucosa zone obtained by optical microscope.

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
