# Peer review of "Recent Advances in Polymer-Based Vaginal Drug Delivery Systems"

_pharmaceutics, 2021, doi:10.3390/pharmaceutics13060884_

Round 1

Reviewer 1 Report

This review by OsmaÅ‚ek et al provides a reasonably comprehensive look at polymer-based vaginal drug delivery systems. Its strong chemical/structural emphasis is a welcome addition and it will be a valuable reference for formulation scientists. For maximum impact and readership, I suggest having a copyeditor polish the manuscript.  My specific comments and suggestions are listed in the attached document.

Line number               Comment

  1. 20 missing a word
  2. 42 refs 2-4 do not describe lactic acid in modern contraceptive gels or creams
  3. 51 missing several key references on HSV-2 and HPV prevention products
  4. 66 delete etc.
  5. 82 how do estrogens influence mucosal tissues?
  6. 95 What is an existing malfunction?
  7. 98 any thoughts/comments on fluid turnover and impact on vaginal drug delivery?
  8. 112                peels off?
  9. 113 has not have
  10. 116 meaning not clear
  11. 119 mucous membrane?
  12. 125 connection and formation
  13. 130 delete etc.
  14. 134 is receptors the correct word?
  15. 146 What about carrageenan?
  16. 173 esterified with methoxy residues sounds odd
  17. 181 methylation is the correct word here
  18. 255 strongly accented
  19. 277 What does low bioavailability mean?
  20. 280 I don’t understand the meaning of this sentence.
  21. 569 What does this sentence mean?
  22. 691 oxidation
  23. 746 dapivirine does not provide contraception
  24. 769 Need to ass Annovera and Ornibel/EluRyng to approved contraceptive rings
  25. 771 ethylene vinyl acetate copolymers = correct name
  26. 772 etonogestrel 0.012 mg and ethinyl estradiol 0.015 mg
  27. 773 it’s per day
  28. 775 is = the correct verb
  29. 782 entire paragraph needs to be revised, and it’s postmenopausal symptoms
  30. 1046 in vivo
  31. 1049 what does CLSM stand for?
  32. 1141 I don’t understand this sentence
  33. 1367 I like the table. How did EFdA and f-fluorouracil jump to the top of the API list?
  34. 1367 Missing anti-HIV APIs like TDF, TAF, griffithsin, MIV-150, and zinc acetate.
  35. 1367 Missing inserts with TAF/EVG (M. Clark) and GRFT (Derby 2018, Lal 2018)

Author Response

This review by Osmałek et al provides a reasonably comprehensive look at polymer-based vaginal drug delivery systems. Its strong chemical/structural emphasis is a welcome addition and it will be a valuable reference for formulation scientists. For maximum impact and readership, I suggest having a copyeditor polish the manuscript. My specific comments and suggestions are listed in the attached document.

Thank you very much for your positive feedback and suggestions for corrections. They have all been corrected in the text

Line number Comment

  1. 20 missing a word - Corrected
  2. 42 refs 2-4 do not describe lactic acid in modern contraceptive gels or creams - New references hav been added concerning the use of lactic acid.
  3. 51 missing several key references on HSV-2 and HPV prevention products – The references have been added to the manuscript.
  4. 66 delete etc. – Removed
  5. 82 how do estrogens influence mucosal tissues? – The sentence has been corrected.
  6. 95 What is an existing malfunction? – The word has been changes to „disease”.
  7. 98 any thoughts/comments on fluid turnover and impact on vaginal drug delivery? – In our oppinion the impact has been clearly indicated in the text.
  8. 112 peels off? – Has been changed to: exfoliate.
  9. 113 has not have - Corrected
  10. 116 meaning not clear – The sentence has been expanded for better clarity.
  11. 119 mucous membrane? – Corrected
  12. 125 connection and formation - Corrected
  13. 130 delete etc. - Deleted
  14. 134 is receptors the correct word? – The sentence has been corrected.
  15. 146 What about carrageenan? – The chapter concerning carrageenans properties has been added to the manuscript.
  16. 173 esterified with methoxy residues sounds odd – In our opinion, the phrase used is correct and does not raise any concerns.
  17. 181 methylation is the correct word here – Corrected
  18. 255 strongly accented - Corrected
  19. 277 What does low bioavailability mean? - The whole sentence has been regarded as misleading and removed from the text.
  20. 280 I don’t understand the meaning of this sentence. The sentence has been modified for better clarity
  21. 569 What does this sentence mean? Physical transition” has been changed to „thickening” for better clarity of the sentence.
  22. 691 oxidation – Corrected
  23. 746 dapivirine does not provide contraception – In the sentence dapivirine is described as antiretroviral agent not contraceptive.
  24. 769 Need to ass Annovera and Ornibel/EluRyng to approved contraceptive rings – The information has been added.
  25. 771 ethylene vinyl acetate copolymers = correct name - Corrected
  26. 772 etonogestrel 0.012 mg and ethinyl estradiol 0.015 mg - Corrected
  27. 773 it’s per day - Corrected
  28. 775 is = the correct verb - Corrected
  29. 782 entire paragraph needs to be revised, and it’s postmenopausal symptoms - The aim of the review was to raise the technological issues of vaginal formulations, all references to vaginal physiology and anatomy as well as the menstrual cycle and postmenopausal period are only general and do not require presenting details.
  30. 1046 in vivo – Corrected.
  31. 1049 what does CLSM stand for? Confocal Laser Scanning Microsopy – Added to the text and abbreviations.
  32. 1141 I don’t understand this sentence – Corrected and explained in the text.
  33. 1367 I like the table. How did EFdA and f-fluorouracil jump to the top of the API list? The table has been entirtely shortened and reorganised also according to other Reviewer’s recommendations.
  34. 1367 Missing anti-HIV APIs like TDF, TAF, griffithsin, MIV-150, and zinc acetate.

We have carefully reanalyzed the table contents. The papers regarding tenofovir disoproxil fumarate (TDF) films and nanoparticles are already mentioned. However, to the best of our knowledge, no scientific report on vaginal formulation with tenofovir alafenamide (TAF) has been published so far. The current studies focus either on subcutaneous administration of long-acting systems (for the reference, please see the following papers: 1. Simpson SM et al., Pharm. Res. 37 (2020) 83, DOI: https://doi.org/10.1007/s11095-020-2777-2; 2. Prathipati PK et al., Pharm. Res. 34 (2017) 2749-2755, DOI: https://doi.org/10.1007/s11095-017-2255-7) or on oral formulations administered as HIV pre-exposure prophylaxis (PrEP), as it was analyzed in well-recognized DISCOVER study (Mayer KH et al., Lancet 396 (2020)239-254; DOI: https://doi.org/10.1016/S0140-6736(20)31065-5). However, none of the prophylactic approaches is referring to the vaginal systems which are the subject of our review. As we mention below, the studies related to vaginal inserts published by M. Clark are only published in the form of review papers and conference proceedings which does not provide a sufficient insight into the methodology of the research projects and the detailed results.

The studies regarding vaginal griffithsin and MIV-150 combined with zinc acetate have been included in the manuscript according to the Reviewer’s remark.

  1. 1367 Missing inserts with TAF/EVG (M. Clark) and GRFT (Derby 2018, Lal 2018)

As far as M. Clark’s works are concerned, vaginal tablets with TAF/EVG are already included in the table. We have carefully considered the other communications regarding vaginal inserts; however, they are only mentioned in one review paper (for the reference, please see: Peet MM et al., Pharmaceutics 11 (2019) 374; DOI: https://doi.org/10.3390/pharmaceutics11080374) and some conference abstracts (1. https://doi.org/10.1089/aid.2016.5000.abstracts, 2. https://www.croiconference.org/abstract/protection-against-vaginal-shiv-infection-insert-containing-taf-and-evg/). We believe that the research area pointed by the Reviewer is interesting and might gain more scientific attention in the future studies. However, as we were not able to get more insight into both projects, we decided to exclude them from the manuscript and focus rather on the studies published in peer-reviewed journals.

The studies on griffithsin authored by Derby and Lal have been added to the manuscript.  

Reviewer 2 Report

The manuscript (pharmaceutics-1232726) entitled "Recent advances in polymer-based vaginal drug delivery systems" provide comprehensive discussion on polymer-based vaginal drug delivery systems. Indeed, manuscript required major revision considering following suggestions to further improve the quality of manuscript. 1. The abstract need to be modified. It should be more specific representing the exact content of the manuscript. Delete the generalized and obvious information, and include current analysis & observation of the authors. 2. The introduction section need to be majorly revised. It is suggested to include the demographic data of different type of vaginal disease and current market size of vaginal product. It is suggested to highlight the challenges and limitations of vaginal drug delivery system and how polymeric-based system helpful to overcome the challenges and current limitations of vaginal drug delivery system. Authors should also highlight the recent development in vaginal product particularly recently available polymeric vaginal product in the market. 3. It is suggested to include an illustration to depict the drug delivery perspective through vaginal route in section 2 of the manuscript. 4. The title of section 3 should be modified as "Polymers utilized in vaginal drug delivery system". It is also advised to include numbering for various polymers discussed in this section. Advantages and limitations of different category of polymers should also be summarized in tabular form for quick understanding to the reader. 5. Classification of products/dosage forms in section 4 are quite confusing and lack rationale basis of classification. It is advised to classify the different vaginal product on particular basis of classification. This section is poorly organized and main weakness of the manuscript. In this section, subsection 4.7. unnecessarily elaborated with all details to increase the bulk of manuscript. In my opinion generalized information which is already available in literature should be deleted and illustrate in a form of flow chart for quick understanding to reader. 6. The table 1 is very lengthy, it is advised to divide this table into multiple number according to type of product/dosage forms and each table inserted into corresponding subsection of the section 4. It is also advised to include one column is this table to incorporate information describing "outcome" of the particular investigation. 7. Summary section should be replaced with "Conclusion and future directions" section with more analysis and observation of the authors related to recent advancement in polymer-based vaginal drug delivery systems. It is also advised to highlight the present limitation of the vaginal products and current research gap this area. 8. It is suggested to recheck whole manuscript thoroughly for the typo mistakes; for example 4.6.3.(. Meth)acrylate polymers (at page 43 line 1027).

Author Response

The manuscript (pharmaceutics-1232726) entitled "Recent advances in polymer-based vaginal drug delivery systems" provide comprehensive discussion on polymer-based vaginal drug delivery systems. Indeed, manuscript required major revision considering following suggestions to further improve the quality of manuscript.

  1. The abstract need to be modified. It should be more specific representing the exact content of the manuscript. Delete the generalized and obvious information, and include current analysis & observation of the authors.

The abstract was entirely modified according to the Reviewer’s remarks.

  1. The introduction section need to be majorly revised. It is suggested to include the demographic data of different type of vaginal disease and current market size of vaginal product. It is suggested to highlight the challenges and limitations of vaginal drug delivery system and how polymeric-based system helpful to overcome the challenges and current limitations of vaginal drug delivery system. Authors should also highlight the recent development in vaginal product particularly recently available polymeric vaginal product in the market.

We have revised the introduction section thoroughly. We have completed information related to the most common vaginal diseases, as well as provided some of the available demographic data to indicate the size of the existing problem. Due to the complex etiology and variety of vaginal diseases in the world, extensive data analysis would significantly exceed the scope of this article, therefore we limited the added text to providing the most important figures relating to the situation in the USA. With regard to the suggestions concerning the analysis of the vaginal formulations market, we believe that due to the large number of marketed formulations, it would require a detailed analysis of data from economic sources, which does not fit into the scope of of this review.

  1. It is suggested to include an illustration to depict the drug delivery perspective through vaginal route in section 2 of the manuscript.

We have added the figure with depicted vaginal route.

  1. The title of section 3 should be modified as "Polymers utilized in vaginal drug delivery system". It is also advised to include numbering for various polymers discussed in this section. Advantages and limitations of different category of polymers should also be summarized in tabular form for quick understanding to the reader.

The title of the section has been changed.

The numbering for the polymers has been added to the text.

In our opinion, due to the various functions that the described polymers can play in vaginal formulations, it would be difficult to clearly indicate their overall advantages and disadvantages. Their role and function significantly depends on the form in which they are applied, as well as the properties of the active substance or additives. We believe that the main and basic properties of the polymers used in the technology of vaginal preparations have been described in the text in a concise manner that will allow even the unexperienced readers to get familia with the topic.

  1. Classification of products/dosage forms in section 4 are quite confusing and lack rationale basis of classification. It is advised to classify the different vaginal product on particular basis of classification. This section is poorly organized and main weakness of the manuscript. In this section, subsection 4.7. unnecessarily elaborated with all details to increase the bulk of manuscript. In my opinion generalized information which is already available in literature should be deleted and illustrate in a form of flow chart for quick understanding to reader.

Section 4 presents an overview of the most frequently investigated and described vaginal dosage forms. All of the mentioned formulation types differ in terms of properties and generally belong to different classes. Therefore, we agree that the whole section might look confusing. In order to highlight some similarities and differences between the described drug delivery systems we have divided them into semisolid dosage forms, solid dosage forms comprising suppositories, tablets, pessaries and vaginal rings, and micro- and nanoparticulate systems. The applied classification was also presented in additional Fig. 2 according to the Reviewer’s remarks.

  1. The table 1 is very lengthy, it is advised to divide this table into multiple number according to type of product/dosage forms and each table inserted into corresponding subsection of the section 4. It is also advised to include one column is this table to incorporate information describing "outcome" of the particular investigation.

Table 1 was reorganized to make it shorter and more reader friendly. To put the column called “outcome” could be useful but it will make the table much more longer. We decided do not put such a column to the table from the beginning because of the number of citied publications and different results. We believe that for readers the information on the active substances and polymers used in the formulation of the drug will be the most important and will get the information for the sources. Also we did not reorganize it in separated tables concerned with the dosage forms because as mentioned above the point was to show what kind of dosage forms were prepared  for given APIs.

  1. Summary section should be replaced with "Conclusion and future directions" section with more analysis and observation of the authors related to recent advancement in polymer-based vaginal drug delivery systems. It is also advised to highlight the present limitation of the vaginal products and current research gap this area.

We have modified and elaborated the summary section (replaced with “Conclusions and future directions” in the corrected version) according to the Reviewer’s remarks.

  1. It is suggested to recheck whole manuscript thoroughly for the typo mistakes; for example 4.6.3.(. Meth)acrylate polymers (at page 43 line 1027).

We have thoroughly checked the manuscript for typo mistakes, grammar and spelling.

Reviewer 3 Report

This is a review article that covers many aspects of vaginal drug delivery.  While the topic is interesting and relevant to the readership of Pharmaceutics, this article is too lengthy and unfocused. It reads more like a textbook chapter than a review. In my opinion, the part of the review that covers the different excipients could be summarized as a table (pointing precisely at the the function of each excipient that makes it particularly relevant to vaginal drug delivery -keeping in mind that these excipients are not exclusively used for vaginal delivery). Overall, the text of the review article could be more appropriately focused on the drug delivery systems, together with the drugs that are being delivered and their intended site of action (local? systemic? exactly what layer of the vaginal wall or the reproductive system are the drugs targeted to?).  In terms of the relative importance being given to drug delivery systems,  there is an imbalance in the importance given to film formulations: films get excessive attention in the review -and I am not sure why that is.  This imbalance makes it seems as if films was the most important drug delivery system which is misleading (each drug delivery system has its function, and they are all important and deserve full consideration, for different particular reasons).  In contrast, there is very little information in terms of the mechanistic relevance of the different formulations/drug delivery strategies in terms of the time-release profile of the API, the localization of the API., and the pharmacokinetics of the API. The authors may want to consider that presenting the time release profile, PK/PD, and intended site of action API for each formulation could be considered as the most mechanistically-relevant information related to the different drug delivery strategies -and the most straightforward way to bring together all the information covered in the article.  

Author Response

This is a review article that covers many aspects of vaginal drug delivery. 

While the topic is interesting and relevant to the readership of Pharmaceutics, this article is too lengthy and unfocused. It reads more like a textbook chapter than a review. In my opinion, the part of the review that covers the different excipients could be summarized as a table (pointing precisely at the the function of each excipient that makes it particularly relevant to vaginal drug delivery -keeping in mind that these excipients are not exclusively used for vaginal delivery).

Undoubtedly, the thematic scope of which we have undertaken to describe ourselves is very extensive. We wanted a holistic approach to issues related to vaginal drugs. Due to the complexity of the topic and the very large number of scientific reports, a thorough review requires providing and describing as many examples as possible. We are convinced that the article will be a valuable source of information, which will undoubtedly translate into a significant number of citations. In the literature, one can often find reviews of greater length than ours. The concept of this work was a comprehensive and comprehensive overview of the current state of knowledge and it was impossible to include all relevant information in a smaller work.

Overall, the text of the review article could be more appropriately focused on the drug delivery systems, together with the drugs that are being delivered and their intended site of action (local? systemic? exactly what layer of the vaginal wall or the reproductive system are the drugs targeted to?). 

We agree that the manuscript is lengthy; however, the presented subject is wide and multidirectional. Macromolecular excipients comprise compounds of different origin, various properties and playing different roles in vaginal drug delivery, as we presented in the review. We believe that even this elaborated text is necessary to summarize the most important information on the latest trends in research area related to vaginal drug delivery systems, even though we did our best to present only the most important data and keep the review concise. We also believe that the subject will be of interest to the readers, especially those who are not very familiar with the subject. However, we think that adding additional information regarding the exact mechanisms of action displayed by the particular active ingredients in the described formulations would additionally increase the size of the manuscript without any significant contribution to the topic of the paper. It must be also emphasized that the results showing the exact localization of the drug released from the designed drug delivery system are not always present in the summarized research papers and provide these data only for some of the mentioned systems would not provide a comprehensive overview of these properties. Moreover, a large part of the described formulations contain well-known pharmaceutical ingredients commonly applied in topical dosage forms and conditions localized in superficial tissues. In such cases the elaborated explanation of pharmacodynamic and pharmacokinetic effects is not necessary.

In terms of the relative importance being given to drug delivery systems,  there is an imbalance in the importance given to film formulations: films get excessive attention in the review -and I am not sure why that is.  This imbalance makes it seems as if films was the most important drug delivery system which is misleading (each drug delivery system has its function, and they are all important and deserve full consideration, for different particular reasons).

In our opinion, and in line with world trends, vaginal films are gaining more and more attention. This is reflected in the growing number of research projects related to them and the number of publications. in our opinion, the chapter is a reliable source of information on vaginal film technology, including methods of their production and evaluation of properties. In our opinion, the chapter is a reliable source of information on vaginal film technology, including methods of their production and evaluation of properties. Therefore, we decided not to shorten the presented text as it could have a negative impact on the quality and value of the article.

In contrast, there is very little information in terms of the mechanistic relevance of the different formulations/drug delivery strategies in terms of the time-release profile of the API, the localization of the API., and the pharmacokinetics of the API. The authors may want to consider that presenting the time release profile, PK/PD, and intended site of action API for each formulation could be considered as the most mechanistically-relevant information related to the different drug delivery strategies -and the most straightforward way to bring together all the information covered in the article. AF

We agree that the detailed analysis of correlation between drug localization, release profiles and pharmacokinetic parameters would be indeed a valuable contribution to the knowledge on vaginal drug delivery and the most useful therapeutic approaches in different conditions. However, most of the available literature reports focusing on the obtaining and characterization of novel vaginal drug delivery system present only very basic data on physicochemical properties and drug release which is most commonly tested with the use of simple in vitro models. More advanced studies include also studies on drug permeation across the excised animal mucosa. The studies showing the in vivo potential of the analyzed systems are scarce and it should be emphasized that most of them are again based on animal models which might not be sufficient to obtain an insight into the efficacy in human subjects. According to the available literature data, there are some significant differences regarding human and porcine vaginal mucosa in terms of drug permeability, even though this tissue is considered as relatively similar to the human one. For the reference, please see the following papers: 1. van Eyk A.D., van der Bijl P. Porcine vaginal mucosa as an in vitro permeability model for human vaginal mucosa. Int. J. Pharm. 305 (2005) 105-111; https://doi.org/10.1016/j.ijpharm.2005.09.002 2. Squier C.A., Mantz M.J., Schlievert P.M., Davis C.C. Porcine vagina ex vivo as a model for studying permeability and pathogenesis in mucosa. J. Pharm. Sci. 97 (2008) 9-21; https://doi.org/10.1002/jps.21077.  We believe that this issue is one of the most important weaknesses of the research focusing on vaginal drug delivery systems and the future studies should aim at the development of novel models allowing for the correlation with the phenomena typical for human anatomy and physiology, as well as better description of the currently applied animal models. Considering the fact that the issue indicated by the Reviewer is of great importance in the design of novel drug delivery systems, we decided to elaborate it in the expanded Summary section.

Round 2

Reviewer 2 Report

In my opinion, the present form of the manuscript is still very lengthy and should be revised considering the following suggestions. 

 1. Table 1 is very lengthy, it is advised to divide this table into multiple numbers according to the type of product/dosage forms, and each divided table should be inserted into the corresponding subsection of section 4.  I do not agree with the author's response to this suggestion. In my opinion, the authors have to consider this suggestion and revised the manuscript accordingly.

  2. Section  4.7. unnecessarily elaborated with all details to increase the bulk of the manuscript and deviated from the scope of the manuscript. The extra emphasis on "films and patches" with all generalized detail which is very obvious information and widely available in literature making confusing and the reader may be lost his interest while reading.  Therefore, in my opinion, this section should be concise, focused, and discussing only the contemporary research & recent development in "films and patches" for vaginal administration.    

Author Response

In my opinion, the present form of the manuscript is still very lengthy and should be revised considering the following suggestions. 

  1. Table 1 is very lengthy, it is advised to divide this table into multiple numbers according to the type of product/dosage forms, and each divided table should be inserted into the corresponding subsection of section 4.  I do not agree with the author's response to this suggestion. In my opinion, the authors have to consider this suggestion and revised the manuscript accordingly.

In our opinion, it is impossible to avoid a large table due to the large amount of references in the article. Regardless of whether we divide the table into several smaller ones or leave it as it is, it will not reduce the volume of the article. We will follow the editor's decision whether the table should be included in the article or the supplementary data.

  1. Section  4.7. unnecessarily elaborated with all details to increase the bulk of the manuscript and deviated from the scope of the manuscript. The extra emphasis on "films and patches" with all generalized detail which is very obvious information and widely available in literature making confusing and the reader may be lost his interest while reading.  Therefore, in my opinion, this section should be concise, focused, and discussing only the contemporary research & recent development in "films and patches" for vaginal administration. 

According to the Reviewer's instructions, we have significantly shortened the mentioned chapter. We left information about the basic properties of films and the latest discoveries in their technology.

Reviewer 3 Report

 The scope, organization and information presented are fine. However, the manuscript is too long and winded, and reads more like a chapter in a textbook than a review article. This is more of an editorial decision, in terms of whether the style and format used by the authors is fit for a review article in Pharmaceutics.

Author Response

The scope, organization and information presented are fine. However, the manuscript is too long and winded, and reads more like a chapter in a textbook than a review article. This is more of an editorial decision, in terms of whether the style and format used by the authors is fit for a review article in Pharmaceutics.

Indeed, the topic we have taken up is very extensive. However, in our opinion, it will be a source of professional information for the readers. Bearing in mind our previous experience with reviews, we are confident that the article will provide a significant number of citations. We also have experience in writing chapters for books, and many of them are well beyond the volume of this article.